# FOCA: Future-Oriented Conditioning for Data-Efficient Vision-Language-Action Adaptation

Duc Minh Nguyen [* 1 2]  Nghiem Tuong Diep [* 1 2]  Binh Gia Nguyen [* 1 2]  Trong-Bao Ho [1]  Doanh Le [2]  Tan Q. Nguyen [1]
Thien-Loc Ha [1]  Nhiem Tran [1]  Bao Thach [1 3]  Nhat X. Tran [1]  Tuan A. Tran [4]  Artur Habuda [5]  Philip Lund Møller [5]
Tran Nguyen Le [5]  Daniel Sonntag [4 6]  Mathias Niepert [7 8]  Khoa D. Doan [2]  Vu Duong [2]  Hung Quoc Ngo [1]  Minh N. Vu [1 2]
Duy M. H. Nguyen [† 4 7 8]  An Thai Le [† 1 2]  Ngo Anh Vien [† 1 2]

## Abstract

Vision–Language–Action (VLA) models enable general-purpose robotic control via large-scale multimodal pretraining, yet their effectiveness under few-shot imitation learning remains limited. We conduct a systematic stress test of state-of-the-art VLA models and show that performance degrades sharply as demonstrations are reduced, revealing a key weakness of existing adaptation strategies. To address this, we introduce FOCA, a future-oriented conditioning framework for data-efficient VLA adaptation. FOCA combines *explicit prediction of task-grounded future interaction* embeddings with *implicit alignment to future goal observations*, enabling long-horizon reasoning in latent space without pixel-level prediction. This formulation naturally *supports action-free* co-training with synthetic videos from video world models and can be interpreted as learning a future-conditioned value-like representation. Extensive experiments demonstrate FOCA achieves 95.7% success with 20 demonstrations on LIBERO, improves 7–12% on RoboCasa, and delivers up to 26% absolute gains on real robots, establishing a new state of the art in few-shot VLA adaptation. Our code is available at this link.

## 1. Introduction

Vision–Language–Action (VLA) models extend large vision–language models to generate robot actions condi-

tioned on visual observations and language instructions, enabling scalable robotic control across tasks, embodiments, and environments. Leveraging large-scale multimodal pretraining, recent VLA systems - spanning web-knowledge transfer to robotics and open generalist robot policies - have demonstrated impressive generalization across diverse manipulation settings and platforms (Zitkovich et al., 2023; Kim et al., 2024; Qu et al., 2025b; Team et al., 2024; Black et al., 2024; Intelligence et al., 2025; Bjorck et al., 2025; Team et al., 2025b;a). Alongside these rapidly advancing model families, recent analyses have begun to clarify key design choices that influence the performance of generalist VLA policies (Liu et al., 2025).

Despite these advances, the data efficiency of VLA models during task adaptation remains insufficiently understood. In practice, collecting task-specific robot demonstrations is costly, and deployment often involves continual distribution shifts - new object instances, changed viewpoints, or minor workspace reconfigurations - that make large demonstration budgets impractical. Yet many evaluations of modern VLAs still adapt with substantially more than a handful of trajectories, leaving open whether pretrained VLA representations reliably translate into *few-shot* adaptability. In particular, the behavior of state-of-the-art VLA models under few-shot imitation learning (e.g., adapting with only 10–30 demonstrations) has not been systematically stress tested. Motivated by this gap, we evaluate strong pretrained VLA models under severely limited demonstration budgets and observe substantial performance degradation, highlighting the need for more data-efficient adaptation mechanisms.

Our key observation is that few-shot adaptation is limited not only by the number of action-labeled trajectories, but also by how much *learning signal* each trajectory provides. Standard behavioral cloning fine-tuning provides one supervised action target per time step; however, each demonstration also contains rich *future observations* that reveal how the scene should evolve when the task is executed successfully. Existing adaptation largely treats demonstrations as independent (observation, action) pairs and does not explicitly leverage the long-horizon structure visible in future frames. These future observations can provide dense, task-aligned supervision about long-horizon task progress, beyond local

[*]Equal contribution . †: Senior Authors (Scientific Advisor: Duy M. H. Nguyen; Core Developers: An Thai Le, Ngo Anh Vien). [1]VinRobotics, Vietnam; [2]Center for AI Research, VinUniversity, Vietnam; [3]University of Utah, USA; [4]German Research Center for Artificial Intelligence (DFKI); [5]Technical University of Denmark, Denmark; [6]University of Oldenburg, Germany; [7]University of Stuttgart, Germany; [8]Max Planck Research School for Intelligent Systems (IMPRS-IS), Germany. Correspondence to: Ngo Anh Vien <v.vienna@vinrobotics.net>.

*Proceedings of the 43rd International Conference on Machine Learning*, Seoul, South Korea. PMLR 306, 2026. Copyright 2026 by the author(s).

action matching. Moreover, future-conditioned learning has long been a principled way to capture long-horizon structure, as in goal-/future-conditioned value functions and successor-style representations (Schaul et al., 2015; Dayan, 1993). We therefore propose FOCA, a *future-oriented conditioning* approach for data-efficient VLA adaptation that explicitly exploits future observations to shape representations and improve long-horizon decision-making.

In contrast to prior approaches that predict pixel-level future frames (Zhao et al., 2025; Hu et al., 2025), FOCA operates entirely in latent embedding space, avoiding the computational cost and redundancy of dense scene reconstruction. Moreover, rather than modeling full scenes, FOCA focuses supervision on task-grounded dynamic regions corresponding to robot–object interactions inferred from the current observation and instruction. We achieve this by introducing a small set of representative tokens that summarize joint vision–language context and are decoded to predict future latent embeddings of interaction regions. This design conditions the policy on anticipated outcomes while remaining invariant to static or task-irrelevant content, aligning with recent advances in self-supervised learning that emphasize prediction in representation space over pixel-space generation (Oord et al., 2018; Assran et al., 2023).

Beyond explicit prediction, FOCA introduces an implicit future-conditioning objective that provides an action-free *auxiliary* supervision signal during adaptation. Instead of predicting future states explicitly, FOCA aligns task-grounded interaction tokens extracted from the current observation with vision embeddings of future goal frames sampled later in the same episode, while contrasting them against embeddings from other tasks. This formulation allows FOCA to directly leverage *synthetic rollouts generated by video world models* (e.g., DreamGen (Jang et al., 2025)) as additional future-structured supervision, without requiring pseudo-actions (Ye et al., 2025) or inverse dynamics (Baker et al., 2022). As a result, the policy can be rapidly adapted to new scenarios by modifying prompts and generating corresponding videos.

Conceptually, the implicit alignment objective encourages the model to encode long-horizon task structure and goal progression directly in representation space. Under geometric future sampling, this objective can be interpreted as estimating goal-conditioned discounted occupancies, a value-like notion of reachability, thereby connecting FOCA to contrastive formulations of goal-conditioned reinforcement learning and successor representations (Watkins & Dayan, 1992; Dayan, 1993). This perspective explains why future-oriented conditioning can provide effective long-horizon supervision during imitation-based adaptation, even though the implicit auxiliary objective itself does not require action labels (Eysenbach et al., 2021).

**Contributions.** We make the following contributions:

- **Few-shot stress test for VLA adaptation.** We provide a systematic evaluation of state-of-the-art VLA models under few-shot imitation learning, revealing substantial degradation when adapting from only 10–30 demonstrations.

- **Future-oriented conditioning for data-efficient adaptation.** We introduce FOCA, which augments VLA adaptation with explicit and implicit future-conditioning objectives in latent space, focusing on task-grounded interaction tokens rather than dense pixel prediction. The implicit objective naturally supports action-free auxiliary supervision from synthetic videos.

- **Strong empirical results across simulation and real robots.** We run experiments across a diverse set of 9 simulated benchmarks, a simulated humanoid and 3 real-robot setups with an Aloha robot arm, observing consistent improvements in few-shot adaptation over strong VLA baselines and adaptation methods.

## 2. Related Works

**VLA Models and Generalist Robot Policies.** Recent work on VLA models unifies perception, language understanding, and control to enable general-purpose robotic manipulation from images and natural language. Existing approaches largely fall into two paradigms. Autoregressive VLAs (e.g., RT-1/2 (Brohan et al., 2022; Zitkovich et al., 2023), OpenVLA (Kim et al., 2024), Gemini (Team et al., 2025b)) generate actions sequentially using large language model backbones, achieving strong generalization through large-scale imitation learning. Diffusion-based VLAs (e.g., $\pi_0$ (Black et al., 2024), Octo (Team et al., 2024), GR00T (Bjorck et al., 2025)) instead synthesize continuous action sequences via diffusion or flow-matching, offering improved robustness and parallel action generation. Despite their success, these models are typically trained and adapted using substantial demonstration budgets, leaving their data efficiency under few-shot adaptation largely unexplored.

FOCA complements these lines of work by addressing data-efficient VLA adaptation. Rather than proposing a new policy architecture, FOCA injects future-oriented learning signals into pretrained VLAs during fine-tuning, enabling effective few-shot transfer across tasks and robot embodiments with significantly fewer demonstrations.

**Few-shot Imitation Learning and Data-Efficient Adaptation.** Adapting a robot policy to new tasks with few demonstrations remains a central challenge in robotics (Duan et al., 2017). Classic behavior cloning is brittle under compounding error, motivating expressive visuomotor imitation models such as Action Chunking with Transformers (ACT) (Zhao et al., 2023) and diffusion-based

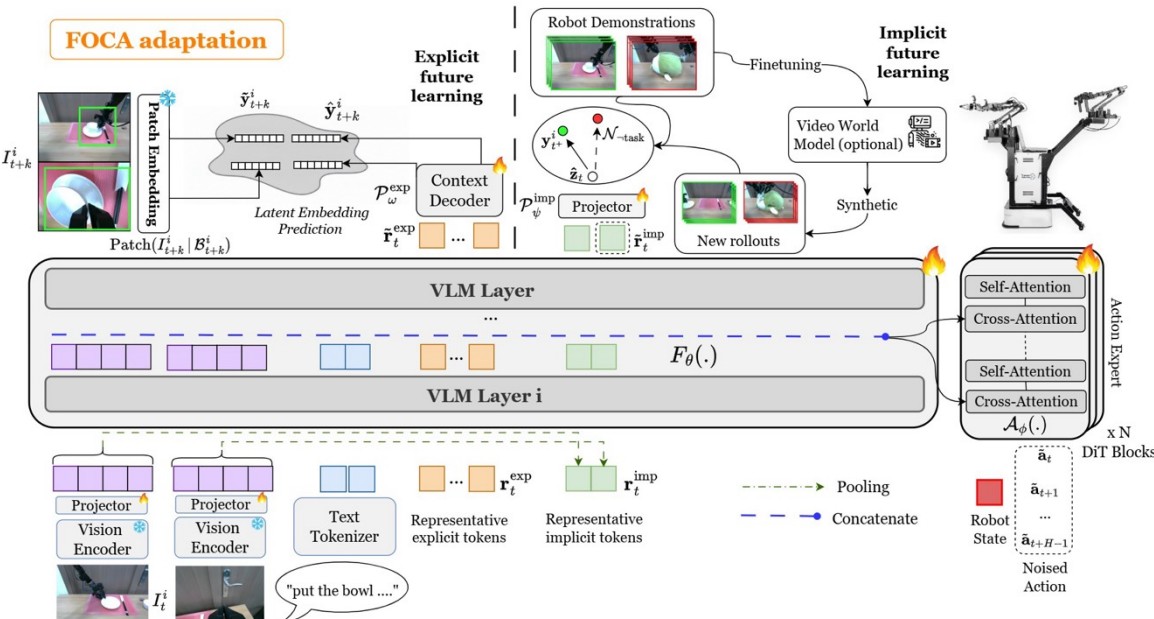

*Figure 1.* FOCA augments VLA adaptation by injecting future-oriented conditioning into a vision–language model $\mathcal{F}_\theta(\cdot)$ and a diffusion-based action policy $\mathcal{A}_\phi(\cdot)$. It jointly learns explicit latent prediction of task-grounded future interaction (bounding boxes $\mathcal{B}_{t+}^i$) embeddings via learnable $\mathbf{r}_t^{\mathrm{exp}}$ tokens and implicit alignment to future goal observations via $\mathbf{r}_t^{\mathrm{imp}}$ tokens, enabling data-efficient, action-free adaptation and seamless integration with synthetic videos from video world models.

policies (Chi et al., 2023). While effective within their training distributions, these models are typically trained from scratch or with limited pretraining, which restricts their ability to generalize to unseen environments and often necessitates substantial task-specific data for adaptation.

At the same time, to mitigate the computational and data cost of adapting pretrained VLA models in low-resource settings (Kim et al., 2024; Mitra et al., 2025), parameter-efficient fine-tuning (PEFT) approaches such as LoRA (Hu et al., 2021) and DoRA (Liu et al., 2024b) have been explored. ControlVLA (Li et al., 2025) in another direction introduces object-centric conditioning via a ControlNet-style interface (Zhang et al., 2023). In contrast, FOCA explicitly incorporates *future-oriented* auxiliary objectives via explicit prediction and implicit alignment to increase the learning signal per demonstration and enable more data-efficient adaptation, including the use of action-free video data.

**Robot Knowledge Prediction and Contrastive Objectives.** Prior VLA models incorporate auxiliary prediction objectives, such as chain-of-thought reasoning (Huang et al., 2025), trajectory generation (Zheng et al., 2025a), or future-state forecasting (Team et al., 2025a), to improve spatial and temporal understanding. In particular, pixel-level future frame prediction has been explored as a form of self-supervision (Seo et al., 2023a; Zhao et al., 2025), but dense image reconstruction is often redundant and difficult to optimize, especially in low-data regimes. FOCA instead predicts future information directly in latent space and focuses

supervision on task-grounded interaction regions, avoiding full-scene reconstruction while preserving control-relevant structure.

Contrastive learning has also been widely used in robotics to improve transfer and generalization, including temporal (Sermanet et al., 2018), multimodal (Rana et al., 2023; Ma et al., 2024), and trajectory-level objectives (Lee et al., 2025). However, most existing methods are designed for CLIP-style models or relatively small policies, and do not readily scale to large pretrained VLA models. FOCA addresses this by introducing a compact set of representative tokens that summarize vision–language context and serve as an efficient interface for future alignment within the VLM. While closely related to FLARE (Zheng et al., 2025b), which applies future alignment during diffusion-policy pretraining, FOCA injects future-oriented objectives directly into the VLM during downstream adaptation and yields additional gains even on GR00-N1.5 (Bjorck et al., 2025), whose diffusion policy already incorporates FLARE-style pretraining. We present in the Appendix more details of the discussion on those related works.

## 3. Methods

### 3.1. Problem Definition

We formulate FOCA as a plug-and-play framework that augments diffusion-based VLA policies, such as $\pi_0$ and GR00-N1.5, for few-shot imitation learning. An overview is shown in Figure 1.

Let $\mathcal{D} = \{(\tau_m, l_m)\}_{m=1}^M$ denote a dataset of episodes paired with task instructions. Each episode $\tau = \{(\mathbf{o}_t, \mathbf{a}_t)\}_{t=1}^T$ consists of multi-view observations $\mathbf{o}_t = [I_t^1, \ldots, I_t^N]$ and action chunks $\mathbf{a}_t = (a_t, \ldots, a_{t+k})$. Given an instruction $l$ and observation $\mathbf{o}_t$, the pretrained vision–language model (VLM) $\mathcal{F}_\theta$ produces a hidden representation:

$$\mathbf{h}_t = \mathcal{F}_\theta(l, \mathbf{o}_t). \tag{1}$$

FOCA augments the VLM by introducing two *types of auxiliary tokens*: an explicit future token set $\mathbf{r}_t^{\text{exp}}$ and an implicit future token set $\mathbf{r}_t^{\text{imp}}$. These tokens are concatenated with $(l, \mathbf{o}_t)$ and jointly processed by the VLM:

$$\mathbf{h}_t, \tilde{\mathbf{r}}_t^{\text{exp}}, \tilde{\mathbf{r}}_t^{\text{imp}} = \mathcal{F}_\theta(l, \mathbf{o}_t, \mathbf{r}_t^{\text{exp}}, \mathbf{r}_t^{\text{imp}}). \tag{2}$$

The token outputs are mapped by lightweight heads. The explicit head $\mathcal{P}_\omega^{\text{exp}}(\cdot)$ produces interaction-focused embeddings from $\tilde{\mathbf{r}}_t^{\text{exp}}$ for predicting future gripper–object interactions, while the implicit head $\mathcal{P}_\psi^{\text{imp}}(\cdot)$ maps $\tilde{\mathbf{r}}_t^{\text{imp}}$ to a contrastive embedding used for alignment with future goal observations across time and episodes.

**Action Generation with Future-Aware Representations.** Given the future-aware representations from the VLM, actions are generated using a denoising diffusion Transformer $\mathcal{A}_\phi(\cdot)$. At diffusion (flow-matching) time $\rho \in [0, 1]$, the policy is conditioned on the robot state $\mathbf{s}_t$, a noised action chunk $\tilde{\mathbf{a}}_{t,\rho}$, and the joint vision–language embeddings $\{\mathbf{h}_t, \tilde{\mathbf{r}}_t^{\text{exp}}, \tilde{\mathbf{r}}_t^{\text{imp}}\}$:

$$\hat{\mathbf{v}}_{t,\rho} = \mathcal{A}_\phi\left(\mathbf{s}_t, \tilde{\mathbf{a}}_{t,\rho}, \rho \,\middle|\, \mathbf{h}_t, \tilde{\mathbf{r}}_t^{\text{exp}}, \tilde{\mathbf{r}}_t^{\text{imp}}\right), \tag{3}$$

where $\hat{\mathbf{v}}_{t,\rho}$ denotes the predicted denoising direction/velocity for the action chunk. At inference time, we start from Gaussian noise and integrate (or iteratively denoise) to obtain the action chunk $\mathbf{a}_t$.

The conditioning mechanism depends on the underlying VLA architecture. GR00T-N1.5 injects the final VLM embedding into $\mathcal{A}_\phi$ via cross-attention at every Transformer layer, whereas $\pi_0$ conditions each diffusion layer using intermediate VLM features through layer-wise cross-attention. We next introduce the future-oriented training objectives.

### 3.2. Explicit Future Latent Prediction

For manipulation, successful execution is largely determined by the future evolution of interactions between the robot's gripper and task-relevant objects specified by the instruction. This motivates forecasting future representations of *object-centric interaction regions* where the gripper engages with language-grounded objects, providing a direct and data-efficient learning signal for action generation.

**Object-Centric Region Construction.** Given a view $I_{t^+}^i$ at time $t^+ > t$ within the same episode and instruction $l$, we use an off-the-shelf grounding model $\mathcal{G}(\cdot)$ to localize task-relevant objects and the robot gripper:

$$\{\mathcal{B}_{t^+,q}^i\}_{q=1}^K = \mathcal{G}(I_{t^+}^i, l),$$

where each $\mathcal{B}_{t^+,q}^i$ is a bounding box corresponding to a language-grounded object or the gripper. We define a single *object-centric interaction region* as the minimal enclosing box:

$$\mathcal{B}_{t^+}^i = \text{Union}\left(\{\mathcal{B}_{t^+,q}^i\}_{q=1}^K\right),$$

which captures the spatial extent of robot–object interactions and serves as the target region for explicit future latent prediction.

**Latent-space Future Interaction Prediction.** At each timestep $t$, we initialize a set of explicit prediction tokens $\mathbf{r}_t^{\text{exp}} \in \mathbb{R}^{n_e \times d}$ as learnable parameters (optionally allocating $n_e/N$ tokens per view) and append them together with the visual tokens from $\mathbf{o}_t$ and language tokens from $l$ as input to the VLM. We map the explicit token outputs $\tilde{\mathbf{r}}_t^{\text{exp}}$ to a prediction space via a lightweight predictor $\mathcal{P}_\omega^{\text{exp}}(\cdot)$ to produce a future region embedding for view $i$ at time $t^+$:

$$\hat{\mathbf{y}}_{t^+}^i = \mathcal{P}_\omega^{\text{exp}}\left(\tilde{\mathbf{r}}_t^{\text{exp}}, \text{PE}(\mathcal{B}_{t^+}^i)\right) \in \mathbb{R}^{N_p \times d_e}, \tag{4}$$

where $N_p$ denotes the number of patch tokens used to represent the interaction region (e.g., via cropping/resizing to a fixed patch grid), $\text{PE}(\cdot)$ is a positional encoding of $\mathcal{B}_{t^+}^i$ (e.g., normalized box coordinates), and $\mathcal{P}_\omega^{\text{exp}}$ can be implemented as a small Transformer.

To form the prediction target, we extract object-centric patch features from $I_{t^+}^i$ restricted to $\mathcal{B}_{t^+}^i$ using a frozen vision encoder $\mathcal{V}(\cdot)$ inside the VLM:

$$\tilde{\mathbf{y}}_{t^+}^i = \text{Patch}(I_{t^+}^i \mid \mathcal{B}_{t^+}^i) \in \mathbb{R}^{N_p \times d_e}. \tag{5}$$

where $\text{Patch}(.)$ denotes a patch layer inside the vision encoder $\mathcal{V}(.)$ (Zhai et al., 2023), extracting image local features and further restricted to tokens inside $\mathcal{B}_{t^+}^i$, so the target captures only interaction-relevant content. Finally, we formulate the predicted embedding to the target in the *latent space across views* as:

$$\mathcal{L}_{\text{exp}} = \mathbb{E}_{t,t^+,i}\left[\left\|\hat{\mathbf{y}}_{t^+}^i - \tilde{\mathbf{y}}_{t^+}^i\right\|_2^2\right]. \tag{6}$$

### 3.3. Implicit Future Alignment

While explicit prediction encourages the model to anticipate *how* object-centric interactions will evolve, the implicit future alignment provides a complementary signal that captures *where* the task is heading at a representation level, encouraging long-horizon task semantics and goal reachability.

Given multi-view observation $\mathbf{o}_t = \{I_t^i\}_{i=1}^N$, we initialize implicit tokens $\mathbf{r}_t^{\text{imp}} \in \mathbb{R}^{N \times d}$ by pooling features from the frozen vision encoder $\mathcal{V}(.)$ followed by another shared learnable projection layer $\mathcal{W}^{\text{imp}}$ as:

$$\mathbf{r}_t^{\text{imp}} = \text{Agg}\left(\{\mathcal{W}^{\text{imp}}(\text{Pool}(\mathcal{V}(I_t^i)))\}_{i=1}^N\right) \tag{7}$$

where $\text{Agg}(.)$ denotes the concatenation operator and

Pool(.) denotes for mean pooling layer.

The output $\tilde{\mathbf{r}}_t^{\text{imp}}$ of $\mathbf{r}_t^{\text{imp}}$ after the VLM model is mapped to a contrastive embedding space via a projection head $\mathcal{P}_\psi^{\text{imp}}(.)$:

$$\{\hat{\mathbf{z}}_t\}_i = \mathcal{P}_\psi^{\text{imp}}\left(\tilde{\mathbf{r}}_t^{\text{imp}}\right) \in \mathbb{R}^{d_e} \tag{8}$$

Similarly to in equation (5), we also define the future goal embeddings at timestep $t^+$ within the same episode as:

$$\mathbf{y}_{t^+}^i = \text{Pool}(\mathcal{V}(I_{t^+}^i)\,|\,\mathcal{B}_{t^+}^i) \in \mathbb{R}^{d_e}, \tag{9}$$

that extract global embedding vision using Pool(.) function and restricted inside the region $\mathcal{B}_{t^+}^i$. Our goal is to make the representative token $\mathbf{r}_{\text{out}}^{\text{imp}}$ encode future-aware representations while remaining discriminative against semantically similar but task-mismatched episodes ($l' \neq l$). Accordingly, we construct the negative set from other minibatch episodes with different task descriptions:

$$\mathcal{N}_{\neg\text{task}} = \left\{\mathbf{y}_{t'}^j\,|\,(\mathbf{o}_{\mathbf{t}'}, l_{m'}) \in \tau_{m'}, m' \neq m, l_{m'} \neq l_m\right\}, \tag{10}$$

where all embeddings $\mathbf{y}_{t'}^j$ are computed using the same object-centric aggregation operator over the corresponding region $\mathcal{B}_{t'}^j$ ($t' > t$). Finally, we optimize the InfoNCE-style loss (Oord et al., 2018):

$$\mathcal{L}_{\text{imp}} = -\mathbb{E}_i \log \frac{\exp\left(s(\mathbf{z}_t^i, \mathbf{y}_{t^+}^i)/\lambda\right)}{\sum_{\mathbf{y} \in \{\mathbf{y}_{t^+}^i\} \cup \mathcal{N}_{\neg\text{task}}} \exp\left(s(\mathbf{z}_t^i, \mathbf{y})/\lambda\right)}, \tag{11}$$

where $s(.;,)$ is cosine similarity, $\lambda$ is a temperature. We sample the future offset $k = t^+ - t$ from a (truncated) geometric distribution, matching the discounted reachability interpretation in Sec. 4.2.

### 3.4. Co-training Future Prediction with Denoising Diffusion Transformer

**Structured Token Isolation between Explicit and Implicit Tasks.** To enforce functional disentanglement between explicit and implicit future modeling, we introduce a *structured token-isolation mask* within the VLM attention layers (Zhao et al., 2025). Specifically, we block direct information exchange between the explicit prediction tokens $\mathbf{r}_t^{\text{exp}}$ and the implicit alignment tokens $\mathbf{r}_t^{\text{imp}}$, while allowing both to attend to shared vision and language tokens: $\text{Attn}(\mathbf{r}_t^{\text{exp}} \leftrightarrow \mathbf{r}_t^{\text{imp}}) = 0$, and $\text{Attn}(\mathbf{r}_t^{\text{exp}} \leftrightarrow \text{vision/text})$, $\text{Attn}(\mathbf{r}_t^{\text{imp}} \leftrightarrow \text{vision/text})$ are unconstrained. The isolation constraint is implemented differently depending on the attention structure of the VLA backbone (bidirectional vs. causal), ensuring the two token groups learn complementary roles without supervisory leakage.

**Co-training across Objectives.** We sample a flow-matching time $\rho \in [0, 1]$ and noise $\epsilon \sim \mathcal{N}(\mathbf{0}, \mathbf{I})$, and form a noised action chunk:

$$\tilde{\mathbf{a}}_{t,\rho} = \rho\,\mathbf{a}_t + (1-\rho)\,\epsilon. \tag{12}$$

For this linear interpolation, the target velocity is $(\mathbf{a}_t - \epsilon)$. We train the diffusion policy to match this target via the flow-matching loss:

$$\mathcal{L}_{\text{fm}} = \mathbb{E}_{t,\rho,\epsilon}\left[\left\|\mathcal{A}_\phi\left(\mathbf{s}_t, \tilde{\mathbf{a}}_{t,\rho}, \rho\,\middle|\,\mathbf{h}_t, \tilde{\mathbf{r}}_t^{\text{exp}}, \tilde{\mathbf{r}}_t^{\text{imp}}\right)\right.\right.$$
$$\left.\left. - (\mathbf{a}_t - \epsilon)\right\|^2\right]. \tag{13}$$

In summary, we co-train three objectives: $\mathcal{L}_{\text{total}} = \mathcal{L}_{\text{fm}} + \mathcal{L}_{\text{exp}} + \mathcal{L}_{\text{imp}}$. For action-free videos (e.g., synthetic rollouts), we omit $\mathcal{L}_{\text{fm}}$ and optimize only $\mathcal{L}_{\text{exp}}$ and $\mathcal{L}_{\text{imp}}$. During inference, the learned tokens are used solely to condition action prediction; no future prediction is performed.

## 4. Analysis of Future-Conditioned Objectives

### 4.1. Action-Free Learning from Synthetic Rollouts

We next highlight a key advantage of FOCA stemming from its implicit future-alignment objective. By relaxing the requirement for precisely localized interaction regions, the objective $\mathcal{L}_{\text{imp}}$ can be directly applied to synthetic videos generated by video world models such as DreamGen (Jang et al., 2025), without relying on pseudo-action labels or inverse-dynamics modeling (Ye et al., 2025). To exploit this property, we adopt a two-stage training strategy: (i) we first fine-tune the video world model using an existing few-shot demonstration, then generate diverse, prompt-conditioned synthetic rollouts to optimize $\mathcal{L}_{\text{imp}}(.)$. This *action-free supervision* shapes FOCA's representations to predict future task goals, enabling effective adaptation with limited real-world data. (ii) We then co-train on real robot demonstrations by jointly optimizing the full objective $\mathcal{L}_{\text{total}}$ This curriculum enables FOCA to absorb future-structured knowledge from large-scale, action-free video while retaining precise control through explicit prediction and action supervision, yielding strong data-efficient adaptation (**Q2. in experiment.**).

### 4.2. Implicit Future Alignment as Goal-Conditioned Discounted Occupancy

FOCA's implicit objective $\mathcal{L}_{\text{imp}}$ (Eq. (11) performs *future-conditioned contrastive learning* between a *context* token from the current input and a *future* interaction/goal embedding sampled later in the same episode. Concretely, let $s_t$ denote the VLA input (multi-view observation plus language) at time $t$. FOCA forms the implicit context token $x_t := z_t$ (Eq. (8)) and a future interaction embedding $g_{t+k} := y_{t+k} = h(s_{t+k})$ (Eq. (9)), where $h(\cdot)$ denotes the region-restricted pooling over dynamic interaction regions. Negatives are drawn from the task-mismatched pool $\mathcal{N}_{\neg\text{task}}$ (Eq. (10)), which we model as a proposal distribution $p_{\text{neg}}(g)$. We show that $\mathcal{L}_{\text{imp}}$ learns a value-like *reachability* score corresponding to discounted goal occupancy under demonstrations (proofs in Appendix F).

*Table 1.* Performance comparison of FOCA vs SOTA VLA models under different data scales.

*(a)* Sensitivity to few-shot imitation learning rate

| Method | Demonstration Ratio | | | | | | | | | | | |
|---|---|---|---|---|---|---|---|---|---|---|---|---|
| | 100% | | | | 40% | | | | 10% | | | |
| | Avg | 10 | Obj. | Spa. | Avg | 10 | Obj. | Spa. | Avg | 10 | Obj. | Spa. |
| $\pi_0$ | 94.6 | 90.0 | 98.2 | 94.6 | 89.9 | 82.0 | 95.2 | 89.6 | 77.6 | 59.0 | 80.6 | 83.4 |
| Groot-N1.5 | 94.6 | **92.8** | 98.4 | 94.4 | 91.4 | 84.5 | 98.9 | 90.6 | 78.2 | 62.6 | 85.7 | 85.7 |
| EO-1 | 94.1 | 91.4 | 96.6 | 89.8 | 91.0 | **88.4** | 96.0 | 86.8 | 82.2 | 65.0 | 89.6 | 83.0 |
| SmolVLA | 92.5 | 82 | 99 | 93 | 90.3 | 80 | 96.0 | 90.0 | 77.3 | 51.3 | 86 | 81.0 |
| **FOCA** | **96.6** | 92.4 | **99.8** | **97.0** | **94.0** | 88.0 | **99.6** | **93.6** | **85.3** | **69.4** | **90.4** | **89.4** |

*(b)* Comparison with general and task-specific PEFT methods for VLA adaptation (average success) in Libero.

| PEFT methods | Demonstration Ratio | | |
|---|---|---|---|
| | 100% | 40% | 10% |
| | Avg | Avg | Avg |
| Control-VLA | 95.6 | 91.3 | 78.4 |
| LoRA ($r=64$) | 94.2 | 90.2 | 78.2 |
| DoRA ($r=64$) | 94.7 | 92 | 78.6 |
| **FOCA** | **96.6** | **94.0** | **85.3** |

*(a)* FOCA vs wide range of VLA models when using full 100% data

| Method | Avg | 10 | Goal | Object | Spatial |
|---|---|---|---|---|---|
| Diff. Policy | 72.4 | 50.5 | 68.3 | 92.5 | 78.3 |
| Octo | 75.1 | 51.1 | 84.6 | 85.7 | 78.9 |
| Open-VLA | 76.5 | 53.7 | 79.2 | 88.4 | 84.7 |
| Spatial-VLA | 78.1 | 55.5 | 78.6 | 89.9 | 88.2 |
| CoT-VLA | 69.0 | 87.6 | 91.6 | 87.5 | 81.1 |
| DreamVLA | 92.6 | 89.5 | 89.5 | 94.0 | **97.5** |
| Groot-N1.0 | 93.9 | 90.6 | 93.0 | 97.6 | 94.4 |
| Groot-N1.5 | 94.6 | **92.8** | 92.8 | 98.4 | 94.4 |
| EO-1 | 94.1 | 91.4 | **98.6** | 96.6 | 89.8 |
| Think-Act | 84.4 | 70.9 | 87.1 | 91.4 | 88.3 |
| SmolVLA | 92.5 | 82.0 | 96.0 | 99.0 | 93.0 |
| $\pi_0$ Fast | 85.5 | 60.2 | 88.6 | 96.8 | 96.4 |
| $\pi_0$ | 94.6 | 90.0 | 95.4 | 98.2 | 94.6 |
| **FOCA (Ours)** | **96.6** | 92.4 | 97.4 | 99.8 | 97.0 |

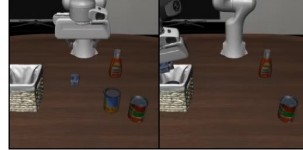
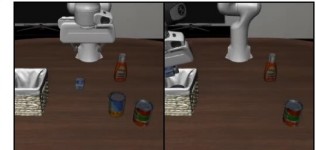

*(b)* put soup and cheese box in basket   *(d)* pick up & place milk in basket

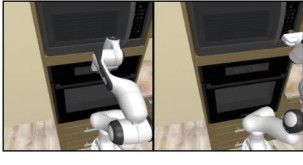
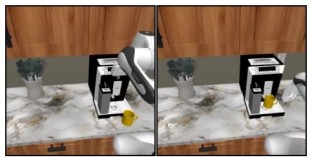

*(c)* RoboCasa-TurnOnMicrowave   *(e)* RoboCasa-CoffeeSetupMug

*Figure 2.* FOCA vs SoTA VLA models on full data scale. 10, Goal, Object, and Spatial denote the Libero-10, Libero-Goal, Libero-Object, and Libero-Spatial benchmark suites, respectively.

**Discounted goal occupancy induced by geometric sampling.** We sample a future offset $k \sim \text{Geom}(1 - \gamma)$ on $\{1, 2, \ldots\}$ and treat $(x_t, g_{t+k})$ as a positive pair. This induces the goal-conditioned *discounted occupancy* under the demonstration distribution:

$$\rho^\mu(g \mid x_t) := (1 - \gamma) \sum_{k=1}^{\infty} \gamma^{k-1} \Pr(g_{t+k} = g \mid x_t), \quad (14)$$

where the probability is taken over trajectories in the offline dataset and the goal extractor $h$.

**Theorem 4.1** (Implicit alignment estimates a goal-conditioned value (log occupancy ratio)). *Assume negatives are sampled from a proposal $p_{\text{neg}}(g)$ independent of $x_t$ (instantiated in FOCA via $\mathcal{N}_{\neg\text{task}}$), and $\mathcal{L}_{\text{imp}}$ is optimized to the population optimum. Then the optimal score satisfies*

$$f_\theta(x_t, g) = \log \rho^\mu(g \mid x_t) - \log p_{\text{neg}}(g) + c(x_t), \quad (15)$$

*for some $c(x_t)$ independent of $g$.*

**Proposition 4.2** (Explicit future prediction estimates an occupancy mean). *Let $\widehat{g}_\theta(x_t)$ denote FOCA's explicit prediction of the future interaction embedding and suppose $\mathcal{L}_{\text{exp}}$ is the squared-error loss in Eq. (6). Under the same geometric sampling as in Eq. (14), any population-optimal predictor satisfies*

$$\widehat{g}^*(x_t) = \mathbb{E}_{g \sim \rho^\mu(\cdot \mid x_t)}[g], \quad (16)$$

*i.e., explicit prediction estimates the conditional mean under the goal-occupancy distribution.*

**Implications for FOCA.** Eq. (15) shows that the similarity score in Eq. (11) acts as a value-like reachability signal in representation space. Because it depends only on observation/language tokens and goal extractor $h$, $\mathcal{L}_{\text{imp}}$ naturally supports action-free training on world-model videos (Sec. 4.1). Meanwhile, Proposition 4.2 clarifies that $\mathcal{L}_{\text{exp}}$ provides a complementary signal by predicting a concrete summary (occupancy mean) of future interaction embeddings, which is directly useful for control. See Appendix F for assumptions, proofs, and more extensions (fixed-horizon futures, robustness bounds, and factorization analyses).

## 5. Experiments

### 5.1. Main results

In this section, we present the main experimental results of FOCA in both simulated and real-world environments (Figure 2-right and Section C Appendix). For details about implementation, other robot setups, we refer readers to the Sections D, E in the Appendix. To systematically evaluate the proposed framework, we design experiments to address the following questions.

**Q1: Data efficiency of FOCA for VLA model adaptation.** We aim to evaluate the data efficiency and robustness of FOCA for VLA adaptation, with an emphasis on few-shot imitation learning. We study its (i) performance across varying demonstration budgets on LIBERO (Liu et al., 2023), corresponding to 10%, 40%, and 100% of

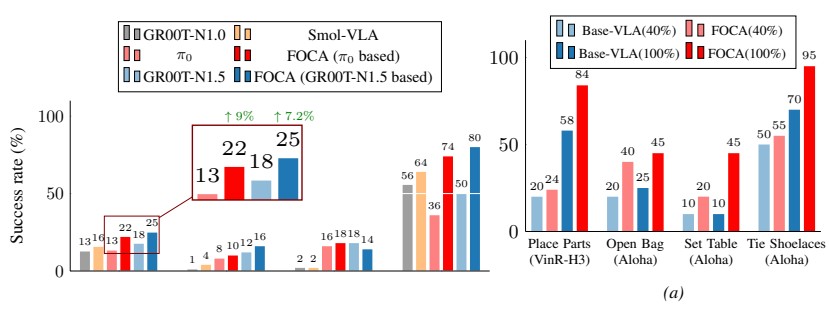

| Data | Method | Avg | 10 | Go. | Obj. | Spa. |
|---|---|---|---|---|---|---|
| 40% | FOCA expli. only | **93.3** | 86.8 | 93.4 | 98.6 | 94.2 |
| | FOCA expli.(full imgs) | 90.9 | 84.0 | 92.6 | 97.2 | 90.0 |
| | FOCA expli.(obj-cen) | 91.4 | 84.4 | 93.6 | 96.2 | 91.4 |
| 10% | FOCAL expli, 8 toks | 79.2 | 60.0 | 89.4 | 83.2 | 84.2 |
| | FOCAL expli, 18 toks | 76.4 | 59.6 | 84.2 | 78.8 | 82.8 |
| | FOCAL expli, 2 toks | 76.6 | 58.6 | 88.0 | 78.6 | 81.2 |

*(b)* Ablation of **explicit future prediction**. We compare latent object-centric prediction (default) with pixel-level prediction on full images and object-centric regions, and study the effect of the number of representative tokens (8 (default), 18 vs. 2).

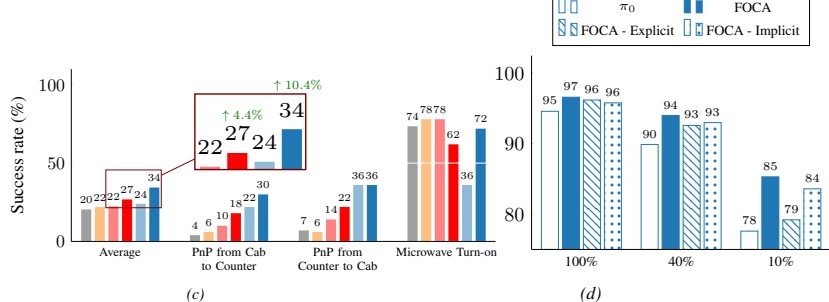

| Method | Avg | 10 | Go. | Obj. | Spa. |
|---|---|---|---|---|---|
| FOCA impli. only - single frame | **83.6** | 65.6 | 91 | 90 | 87.6 |
| FOCA impli. only - multi-frames | 83.3 | 64.8 | 92 | 89.5 | 86.9 |
| FOCA impli. with full images | 81.7 | 62.4 | 90.8 | 87.6 | 85.8 |
| Contrast - image-text vs. robot state using output VLM | 78.2 | 59.4 | 86.2 | 85.4 | 81.6 |
| Contrast - multi-frames vs. text | 78.3 | 59 | 87 | 83.2 | 84 |

*(e)* Ablation study on **implicit future prediction** (on 10% data Libero) with: (i) (default) single future frame alignment ($t+10$), (ii) using multi-frames in future ($t+10,t+20,t+30$) (second row), (iii) using full embedding images rather than object centric (third row), (iv) other contrastive strategies developed for small-scale models (Rana et al., 2023; Ma et al., 2024).

*Figure 3.* (a) FOCA's performance on four real-robot tasks using a humanoid (VinR-H3, a simulated task) (GR00T N1.5 VLA, evaluation on 50 trials), and three other with bi-arm Aloha robot using $\pi_0$ VLA; (b) Ablation study on explicit future prediction ; (c) FOCA's generalization performance across $\pi_0$ and GR00T N1.5 on Robocasa, with 30 demos (**top**) and 100 demos (**bottom**) on most 5 challenge tasks; (d) $\pi_0$ vs FOCA, FOCA w explicit, and FOCA w implicit at 100%, 40% and 10%; (b) and (e) be ablation study on explicit and implicit future prediction.

*Table 2.* Performance comparison between FOCA variants and pseudo-actions learned via inverse generative modeling (IGM) from DreamGen-generated synthetic videos.

| Method | Demonstration Ratio | | | | | | | | | | | | | | |
|---|---|---|---|---|---|---|---|---|---|---|---|---|---|---|---|
| | 100% | | | | | 40% | | | | | 10% | | | | |
| | Avg | 10 | Go. | Obj. | Spa. | Avg | 10 | Go. | Obj. | Spa. | Avg | 10 | Go. | Obj. | Spa. |
| $\pi_0$ | 94.6 | 90.0 | 95.4 | 98.2 | 94.6 | 89.9 | 82.0 | 92.6 | 95.2 | 89.6 | 77.6 | 59.0 | 87.4 | 80.6 | 83.4 |
| IGM | 94.3 | 90.0 | 95.2 | 98.2 | 93.6 | 90.2 | 82.8 | 95.6 | 94.8 | 87.8 | 76.8 | 61.8 | 84.0 | 79.8 | 81.4 |
| **FOCA(Imp.)** | 95.8 | 91.0 | 97.2 | 98.4 | 96.6 | 93.0 | 87.0 | 94.4 | 97.6 | 92.9 | 83.6 | 65.6 | 91.0 | 90.0 | 87.6 |
| **FOCA(D.G.)** | **96.7** | **92.6** | **96.8** | **99.2** | **98.0** | **95.7** | **89.4** | **96.6** | **98.6** | **98.0** | **86.4** | **73.2** | **91.4** | **91.8** | **89.2** |

the training data (approximately 5, 20, and approximately 43 demonstrations per task, respectively); (ii) compare it against parameter-efficient fine-tuning methods, and assess (iii) its competitiveness with state-of-the-art VLA models under scarce and full supervision.

→ **Baselines and evaluation protocol.** We integrate FOCA into the $\pi_0$ (Black et al., 2024) vision–language–action (VLA) model and compare against a diverse set of strong pretrained VLA baselines adapted via standard imitation learning, including GR00T-N1.5 (Bjorck et al., 2025), EO-1 (Qu et al., 2025a), and SMOLVLA (Shukor et al., 2025). These models represent state-of-the-art (SOTA) generalist policies across diverse architectural scales, with SMOLVLA demonstrating competitive generalization despite its smaller backbone. All methods are evaluated under identical data splits, training budgets, and evaluation protocols.

To disentangle the effect of future-oriented conditioning from parameter efficiency, we additionally compare against

representative parameter-efficient fine-tuning (PEFT) methods applied to the same $\pi_0$ backbone, including LORA (Hu et al., 2021), DORA (Liu et al., 2024b), and CONTROL-VLA (Li et al., 2025). These baselines capture a broad spectrum of recent PEFT strategies designed to stabilize or regularize VLA adaptation under limited supervision.

Finally, under the full-data setting (100%), we benchmark FOCA against a wide range of state-of-the-art generalized robot policies and VLA systems reported in the literature, including DIFFUSION POLICY (Chi et al., 2023), OCTO (Team et al., 2024), OPEN-VLA (Kim et al., 2024), SPATIAL-VLA (Qu et al., 2025b), COT-VLA (Zhao et al., 2025), DREAMVLA (Zhang et al., 2025), THINK-ACT (Huang et al., 2025), and $\pi_0$-fast (Pertsch et al., 2025). Among these, THINK-ACT, COT-VLA, and DREAMVLA also incorporate future prediction objectives, but operate at the text reasoning or pixel level, and do not model task-grounded interaction representations in latent space. Reported results are taken from the original publications, except for EO-1, which we reimplement due to prohibitive default training time (40 hours), while FOCA and other methods converge within 20 hours under our setup.

→ **Result.** As shown in Table 1 and Fig. 2(a), FOCA consistently improves data efficiency for VLA adaptation. With only **40% of demonstrations**, **FOCA** achieves **94.0%** success on LIBERO, closely matching $\pi_0$ trained with **100% data (94.6%)**, while in the 10% regime it substantially re-

duces the few-shot gap, improving success **from 77.6% to 85.3%** and outperforming PEFT methods, which plateau around ~78%. FOCA also attains state-of-the-art performance under full supervision. These results suggest that *few-shot VLA adaptation is limited more by the quality of supervision than by model capacity*; FOCA addresses this by using future observations to provide richer learning signals that improve long-horizon behavior.

**Q2: Leveraging synthetic videos without action supervision.** We investigate whether FOCA effectively leverages synthetic rollouts from video world models without access to action labels, and how this compares to approaches that rely on pseudo-actions inferred from video.

→ **Baseline setup.** For each data regime (10%, 40%, and 100%), we first fine-tune the open-world model Dream-Gen (Jang et al., 2025) on the corresponding real demonstrations. Using the resulting checkpoint, we generate synthetic task-consistent videos conditioned on task descriptions and initial frames from training episodes. These videos are then used in two ways.

- Inverse generative modeling (IGM) (Baker et al., 2022): we train the IGM model on the synthetic videos to infer pseudo-actions, which are subsequently used to fine-tune the $\pi_0$ VLA model.

- FOCA variants: we consider two setups. (a) FOCA + IMPLICIT co-trains the VLA model using action supervision on real data together with FOCA's implicit future-alignment objective *(we disable explicit future prediction)*. (b) FOCA + DREAMGEN first pretrains the VLA model using FOCA's implicit objective on DreamGen-generated videos only, then continues training as in FOCA + Implicit using real demonstrations.

→ **Results.** As shown in Table 2, pseudo-action–based **IGM yields no consistent improvement** over $\pi_0$, matching performance only with full data and degrading under limited supervision. In contrast, **FOCA consistently improves** performance across all regimes: even with implicit alignment alone, FOCA outperforms both $\pi_0$ and IGM, and incorporating synthetic videos further amplifies these gains at **40% data**, **FOCA + DreamGen (95.7%)** surpasses $\pi_0$ trained with **100% data (94.6%)**. These results indicate that pseudo-action inference is brittle due to noise in recovered actions, while FOCA's **future-oriented conditioning** leverages synthetic videos as representation-level supervision, remaining robust even when generated rollouts are imperfect. We present additional details about DreamGen's synthetic rollouts in the Appendix.

**Q3: How does FOCA generalize across different VLA architectures and different real robotic embodiments?**

**FOCA on GR00T N1.5.** To assess whether FOCA generalizes beyond a single backbone, we integrate it with two architecturally distinct VLA models, $\pi_0$ and GR00T-N1.5, and evaluate on the ROBOCASA benchmark under both 30- and 100-demonstration regimes across five challenging manipulation tasks. We additionally compare against GR00T-N1.0 and SMOLVLA. As shown in Fig. 3c, FOCA consistently improves performance across all tasks and data scales, yielding **gains of 4.4 - 12%** on $\pi_0$ and **7.2 - 10.4%** on **GR00T-N1.5**. These results demonstrate that FOCA generalizes effectively across VLA architectures. Full results are provided in the appendix (Figure 10).

**FOCA in real-world setups.** We further evaluate FOCA in three real-robot setups spanning two distinct robotic platforms - bimanual ALOHA ROBOT (Aldaco et al., 2024) and a simulated HUMANOID PLATFORM - and diverse manipulation tasks: *Place Parts, Open Bag, Set Table, and Tie Shoelaces*. The details for robot setup are presented in detail in Sec. E, with task examples that can be visualized in Sec. C, Appendix. The results are summarized in Figure 3a.

Across real-world robot platforms, FOCA consistently improves performance over base VLA models under both limited and full supervision. On the simulated **humanoid**, FOCA boosts GR00T-N1.5 from **58% to 84%** success rate on the Place Part task, demonstrating strong transfer to an industrial-scale robot with distinct kinematics and sensing.

On the **Aloha bimanual robot**, FOCA yields substantial gains across challenging household tasks using $\pi_0$ as the backbone. Notably, FOCA doubles success on *Open Bag* under limited data *(20% → 40%)* and improves full-data performance to 45%. On the highly dexterous Tie Shoelaces task, it reaches 95% success. Together, these results demonstrate that FOCA generalizes robustly across robot embodiments, task types, and real-world deployment conditions.

### 5.2. Ablation study

**Contribution of explicit and implicit components.** We ablate FOCA's explicit and implicit modules on LIBERO across different data regimes (Fig. 3d). Full FOCA consistently outperforms both ablations and the base $\pi_0$, with the gap widening as data decreases; at 10% data, FOCA achieves 85.3%, substantially outperforming either component alone. This confirms that explicit interaction-centric prediction and implicit goal-level alignment provide complementary signals that are critical in low-data settings.

**Explicit and implicit ablations.** We ablate FOCA's explicit and implicit future-conditioning designs under moderate and low-data regimes. For explicit one (Fig.3(b)), operating in a *latent, object-centric space* consistently *outperforms pixel-level alternatives*, showing that structured representations provide more robust supervision by suppressing task-irrelevant visual variability; performance is further sensitive to the number of representative tokens, with a moderate token count yielding the best results. For implicit

*Table 3.* Ablation study on token initialization strategies for implicit and explicit tokens in FOCA on LIBERO at 40% data scale.

| Method | 10 | Go. | Obj. | Spa. | Avg |
|---|---|---|---|---|---|
| $\pi_0$ Baseline | 82 | 92.6 | 95.2 | 89.6 | 89.85 |
| FOCA (ada. impli., rand. expli.) | 88 | 94.6 | 99.6 | 93.6 | 93.95 |
| FOCA (both random) | 84.2 | 92.6 | 97.8 | 92.2 | 91.7 |
| **FOCA (both adaptive)** | **87.8** | **96** | **98.4** | **95** | **94.3** |

*Table 4.* Ablation study on token initialization strategies for implicit and explicit tokens in FOCA on LIBERO at 100% data scale.

| Method | 10 | Go. | Obj. | Spa. | Avg |
|---|---|---|---|---|---|
| $\pi_0$ Baseline | 90 | 95.4 | 98.2 | 94.6 | 94.6 |
| FOCA (ada. impli., rand. expli.) | 92.4 | **97.4** | **99.8** | **97** | 96.6 |
| **FOCA ((both adaptive)** | **92.8** | 97.2 | 99.6 | **98.2** | **97** |

alignment (Fig. 3(e)), focusing on a single, well-aligned future frame with object-centric representations yields strong performance, while extending alignment to *multiple future frames achieves comparable results*. In contrast, *alternative contrastive objectives are less effective*. Together, these ablations show that our design choices are critical for extracting long-horizon learning signals in few-shot settings.

**Token initialization strategies for implicit and explicit tokens.** We also compare four variants: the $\pi_0$ baseline, static (random) initialization for both token types, dynamic (adaptive) initialization for implicit tokens only, and dynamic initialization for both token types. Results from Table 3 and 4 show that dynamic initialization consistently improves performance, with the fully adaptive variant achieving the best average score, validating the importance of grounding both token types in visual features for effective future-oriented conditioning.

**Comparison with re-implemented future-oriented baselines.** We also compare FOCA with representative future-oriented baselines, including DreamVLA-style (Zhang et al., 2025) and FLARE (Zheng et al., 2025b), under the same policy backbones and fine-tuning settings, as shown in Tables 5, 6, and 7. For fair comparison, DreamVLA-style dynamic future conditioning is integrated into $\pi_0$, while FLARE is re-implemented following its original design due to the absence of publicly available fine-tuning code. Results show that future-oriented conditioning generally improves over the corresponding baselines across most tasks and benchmarks. However, FOCA consistently achieves the best overall performance in all settings. On LIBERO with $\pi_0$ at 40% data scale, FOCA improves the average score from 92.1 (DreamVLA) and 91.0 (FLARE) to 94.0. Similarly, at 100% data scale, FOCA outperforms DreamVLA-style conditioning by +1.0 average score. On RoboCasa with Gr00t-N1.5, FOCA further achieves 34.4 average success rate compared to 28.4 for FLARE. These results demonstrate the effectiveness and robustness of FOCA across different future-oriented baselines, policy architectures, and

*Table 5.* Comparison of $\pi_0$ fine-tuned with DreamVLA-style future conditioning, FLARE, and FOCA on LIBERO at 40% data scale.

| Method | 10 | Go. | Obj. | Spa. | Avg |
|---|---|---|---|---|---|
| $\pi_0$ Baseline | 82 | 92.6 | 95.2 | 89.6 | 89.85 |
| $\pi_0$ + DreamVLA | 84.8 | 93.2 | 96.4 | 94 | 92.1 |
| $\pi_0$ + FLARE | 83.6 | 93 | 96.8 | 90.6 | 91 |
| **FOCA** | **88** | **94.6** | **99.6** | **93.6** | **94** |

*Table 6.* Comparison of $\pi_0$ fine-tuned with DreamVLA-style future conditioning and FOCA on LIBERO at 100% data scale.

| Method | 10 | Go. | Obj. | Spa. | Avg |
|---|---|---|---|---|---|
| $\pi_0$ Baseline | 90 | 95.4 | 98.2 | 94.6 | 94.6 |
| $\pi_0$ + DreamVLA | 90.8 | 96.4 | 98.8 | 96.2 | 95.6 |
| **FOCA** | **92.4** | **97.4** | **99.8** | **97** | **96.6** |

robotic manipulation benchmarks.

*Table 7.* Comparison of Gr00t-N1.5 fine-tuned with FLARE and FOCA on RoboCasa at 100 demos.

| Method | Pick-and-Place | | Coffee Setup | Off Stove | On MW | Avg |
|---|---|---|---|---|---|---|
| | Cab→Counter | Counter→Cab | | | | |
| Gr00t-N1.5 Baseline | 28 | 40 | 12 | 14 | 36 | 26 |
| Gr00t-N1.5 + FLARE | 24 | 34 | 20 | 14 | 50 | 28.4 |
| **Gr00t-N1.5 + FOCA** | **30** | **36** | **10** | **24** | **72** | **34.4** |

**Contribution of DreamGen on FOCA on table setup tasks using bimanual Aloha robot.** We ablate the effectiveness of DreamGen on FOCA when apply to real-robot (bimanual Aloha) on table set up tasks. Our models are trained with the $\pi_0$ VLA architecture using either 40% (120 demonstrations) or 100% (300 demonstrations) of the dataset. As shown in Table 8, incorporating FOCA improves performance over $\pi_0$, and combining FOCA with DreamGen yields the largest gains.

*Table 8.* Comparison of task success rates on real-robot.

| | $\pi_0$ | FOCA ($\pi_0$) | **FOCA + DreamGen** |
|---|---|---|---|
| 40% | 10% | 20% | 55% |
| 100% | 10% | 45% | 60% |

## 6. Conclusions

We show that few-shot VLA adaptation is limited by the learning signal extracted per demonstration. FOCA addresses this by combining explicit and implicit future supervision in the latent space to enable data-efficient, action-free adaptation of pretrained VLA models. Extensive experiments across simulation and real robots demonstrate consistent gains, robustness to imperfect synthetic rollouts, and competitive generalization across architectures and embodiments, positioning future-conditioned representation learning as a promising route toward scalable robotic adaptation.

## Impact Statement

This paper presents work whose goal is to advance the field of Data Efficiency in Robotics and Vision-Language-Action models. There are many potential societal consequences of our work, none which we feel must be specifically highlighted here.

## Acknowledgment

Duy M. H. Nguyen and M. Niepert acknowledge funding by Deutsche Forschungsgemeinschaft (DFG, German Research Foundation) under Germany's Excellence Strategy - EXC 2075 – 390740016 and the International Max Planck Research School for Intelligent Systems (IMPRS-IS). Tuan Anh Tran, Duy M. H. Nguyen, and Daniel Sonntag are also supported by the No-IDLE project (BMFTR, 16IW23002), the MASTER project (EU, 101093079), and the Endowed Chair of Applied Artificial Intelligence, Oldenburg University.

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

SUPPLEMENTARY MATERIAL FOR
"FOCA: FUTURE-ORIENTED CONDITIONING FOR DATA-EFFICIENT
VISION-LANGUAGE-ACTION ADAPTATION"

## Contents

# A. Limitations and Future Works

**Limitations.** While FOCA improves few-shot adaptation by injecting future-oriented supervision, it does not explicitly model long-term symbolic reasoning or task decomposition. In complex, multi-stage tasks that require remembering completed subtasks and planning several steps ahead (e.g., scenarios with multiple similar objects such as bartender-style setups), the lack of an explicit memory or chain-of-thought mechanism can lead to ambiguity or confusion. FOCA captures long-horizon structure implicitly in representation space, but it does not generate intermediate reasoning traces or maintain a persistent task state, which may limit performance on tasks requiring explicit temporal abstraction and bookkeeping.

**Future work.** An important direction is to extend FOCA with explicit reasoning and memory components, such as lightweight task-state memories or structured chain-of-thought generation, to better handle long-horizon, compositional tasks. Another promising avenue is to investigate FOCA's applicability to autoregressive VLA architectures, including EO-1 and CoT-VLA, where future-oriented conditioning could be integrated at the token-generation level rather than within diffusion policies. More broadly, combining FOCA with structured reasoning, memory, and autoregressive control may offer a unified framework that bridges representation-level future conditioning with explicit planning and decision-making.

# B. Additional Related Works.

**Synthetic Data and Viewpoint Augmentation.** A complementary line of work improves data efficiency by augmenting demonstrations with additional observations, including synthesizing trajectories under novel robot embodiments or camera viewpoints to improve robustness and transfer (e.g., RoVi-Aug (Chen et al., 2024)). Relatedly, multi-view representation learning exploits auxiliary viewpoints during training to improve downstream control without requiring camera calibration at deployment (Seo et al., 2023b). More broadly, video and world models can generate plausible future rollouts that serve as auxiliary supervision or evaluation scaffolding, including robotics-oriented generators such as DreamGen (Jang et al., 2025) and large-scale world-model platforms such as Cosmos (Agarwal et al., 2025a; Ali et al., 2025). FOCA is compatible with these approaches but is motivated by a distinct observation: synthetic videos can provide useful future supervision even when action labels are unavailable.

Rather than introducing a new VLA backbone or pretraining recipe, FOCA operates as a fine-tuning–time adaptation mechanism that can be layered on top of existing VLA models and adaptation strategies, including parameter-efficient updates and object-centric conditioning (Hu et al., 2021; Liu et al., 2024b; Li et al., 2025; Sridhar et al., 2025). By injecting future-oriented objectives in latent space over task-grounded interaction tokens, FOCA introduces long-horizon structure while remaining lightweight and effective in few-shot regimes. Crucially, its implicit future alignment naturally extends to action-free videos generated by world models, enabling the use of synthetic rollouts without inverse dynamics or pseudo-action labeling (Agarwal et al., 2025a). Overall, FOCA bridges large-scale VLA pretraining and data-efficient downstream adaptation in a principled and modular manner.

**Future Prediction in VLA models** Recent VLA models incorporate auxiliary prediction objectives to improve spatial reasoning, including chain-of-thought prediction, trajectory generation, visual question answering, and future state forecasting (Zheng et al., 2025a; Qu et al., 2025a; Team et al., 2025a). Among these directions, pixel-level future frame prediction has emerged as a particularly appealing form of self-supervision (Seo et al., 2023a; Zhao et al., 2025), but dense image reconstruction is often redundant and difficult to optimize, especially in low-data regimes. In contrast, FOCA predicts future information directly in latent embedding space and restricts supervision to task-grounded dynamic regions capturing gripper–object interactions, which can be efficiently identified using off-the-shelf visual grounding models (e.g., DINO (Liu et al., 2024a)), thereby avoiding full-scene reconstruction while preserving the control-relevant inductive bias needed for effective robot adaptation.

**Contrastive Learning in Robotic Manipulation** Contrastive learning has been widely used to improve transfer and generalization in robotic learning, including time-contrastive objectives across multi-view observations (Sermanet et al., 2018), multimodal alignment between vision, language, and actions (Rana et al., 2023; Ma et al., 2024), and trajectory-level contrastive losses in latent diffusion spaces (Lee et al., 2025). However, these methods are largely developed for CLIP-based models or relatively small policies such as ACT and Diffusion Policy (Chi et al., 2023). Directly applying them to large pretrained VLA models is challenging due to model scale and the domain gap between pretraining and downstream control. To address this, FOCA introduces a small set of representative tokens that summarize vision–language information via attention within the VLM and serve as a compact interface for contrastive future alignment. The most closely related work is

FLARE (Zheng et al., 2025b), which integrates future alignment during large-scale *diffusion* pretraining; in contrast, FOCA applies future-oriented objectives directly *within the VLM modules* during downstream adaptation. Furthermore, we show that FOCA further improves GR00T-N1.5 (Bjorck et al., 2025) performance, whose diffusion policy is already pretrained with FLARE, demonstrating that FOCA provides additional gains beyond diffusion-level pretraining alone.

# C. Visualizations

This section provides qualitative visualizations to complement the quantitative results presented in the paper. We include representative examples from both simulated and real-world environments to illustrate task diversity, long-horizon structure, and failure modes addressed by FOCA.

## C.1. Rollout Visualizations on Simulated and Real-World Robots

Figures 4 and 5 show example task executions from the LIBERO and RoboCasa benchmarks, respectively. These environments span a wide range of manipulation challenges, including object placement, spatial reasoning, and multi-step goal completion under varying initial conditions. Figures 6 and 7 present example rollouts collected on two real-world robotic platforms: the bimanual ALOHA robot and a simulated humanoid robot. These figures illustrate the visual complexity and embodiment-specific challenges present in real-world settings, including partial observability, viewpoint variation, and contact-rich manipulation.

pick up the black bowl on the ramekin and place it on the plate

pick up the milk and place it in the basket

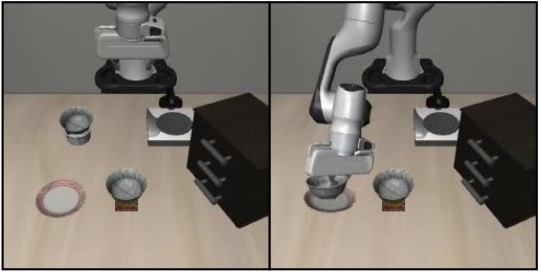
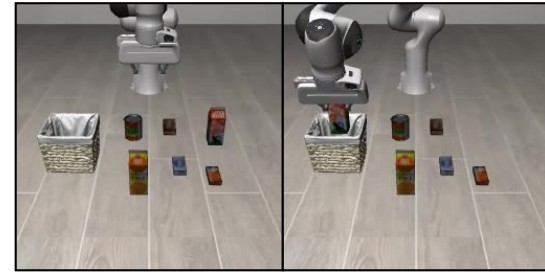

put the bowl on the stove

put both the alphabet soup and the cream cheese box in the basket

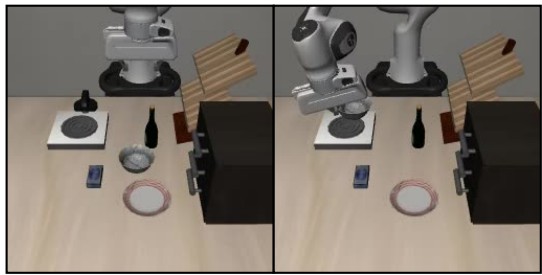
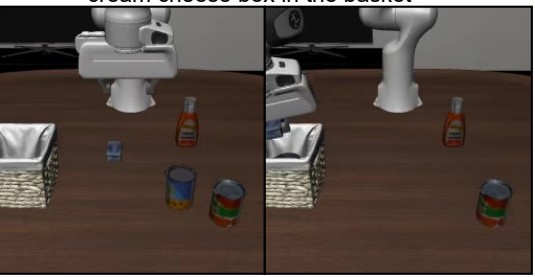

*Figure 4.* **Example rollouts from LIBERO simulation.**

## C.2. Failure correction via future conditioning

To better understand the behavioral differences induced by future-oriented conditioning, we visualize continuous frame sequences for representative failure cases (Figure. 8). In particular, we compare baseline VLA policies that fail to complete the task with FOCA-enhanced policies that successfully recover and reach the goal under the same initial conditions. These examples qualitatively demonstrate that FOCA improves long-horizon consistency and prevents early commitment to suboptimal interaction trajectories, consistent with its future-conditioned design.

CabToCounter

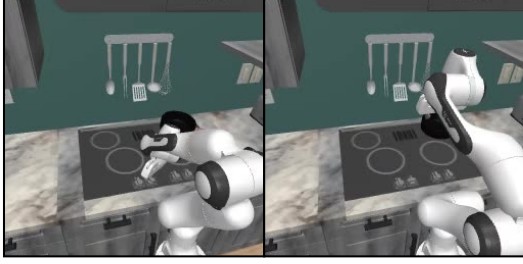

TurnOffStove

TurnOnMicrowave

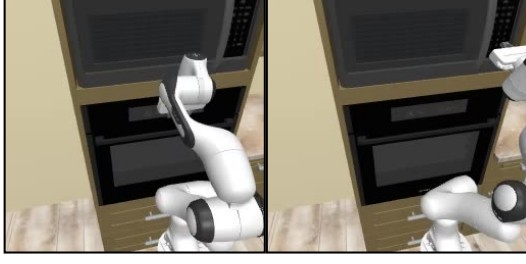

CoffeeSetupMug

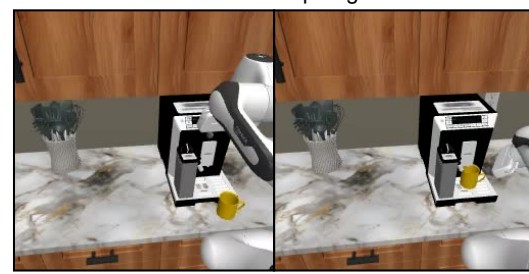

*Figure 5.* **Example rollouts from ROBOCASA simulation.**

### C.3. DreamGen hallucination analysis

Figure 9 illustrates sequences generated by the DreamGen video world model under different training regimes. When trained with limited demonstrations (40%), the generated videos exhibit hallucinations. In contrast, DreamGen trained with a larger few-shot imitation ratio (100%) produces more coherent and physically plausible future trajectories.

## D. Implementation Details

### D.1. Architecture designs

We evaluate our proposed method, **FOCA** (explicit module is demonstrated in Figure 11), on two public simulation datasets - *LIBERO* and *RoboCasa* - as well as two distinct robotic platforms - bimanual ALOHA ROBOT and simulated HUMANOID PLATFORM, version VR-H3 designed by VinRobotics, using Inspire's 6-DOF hands. We apply FOCA to $\pi_0$ on both public datasets and ALOHA ROBOT tasks. For GR00T-N1.5, FOCA is evaluated on RoboCasa and the simulated HUMANOID PLATFORM. For the $\pi_0$ architecture, the batch size is set to 18 for LIBERO and 12 for RoboCasa. The learning rate and maximum number of training steps are fixed to $1 \times 10^{-4}$ and 100k steps, respectively, for both datasets. For the GR00T-N1.5 architecture on RoboCasa, the batch size, learning rate, and maximum number of training steps are set to 32, $1 \times 10^{-4}$, and 60k steps, respectively. Note that both the baseline methods and our proposed FOCA use identical hyperparameter settings to ensure a fair comparison. The training process can be early-stopped once the loss curve has sufficiently converged.

For the decoder in the explicit module, we employ two Transformer blocks with 16 attention heads, followed by a linear layer that projects the outputs into the visual *local* feature space. We set the query length to 4 for each observation and use a shared decoder across all observations, with the explicit loss weighted by 0.1. For the implicit module, we use a single linear layer to project the implicit tokens produced by the VLM into the visual *global* feature space. The number of parameters for each model before and after applying FOCA is reported in Table 9.

### D.2. Task-Grounded Object-Centric Representation Generation across Timesteps

To construct the the *Object-of-interest regions* used in our experiments, we derive spatial supervision differently for simulation datsets and realworld data.

For the simulation environments include LIBERO and RoboCasa, we extract segmentation masks from simulator. From

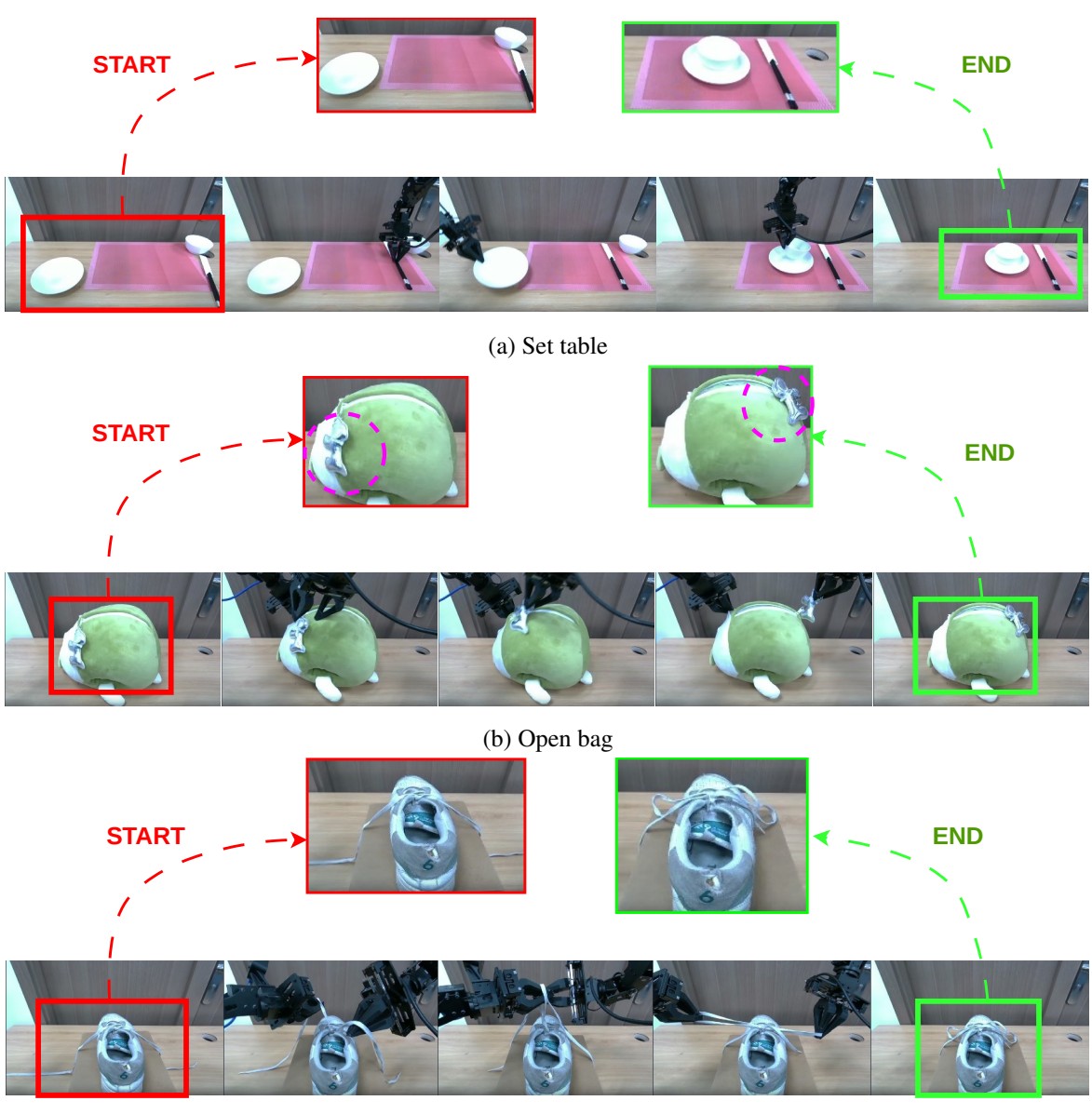

(a) Set table

(b) Open bag

(c) Tie shoelaces

Figure 6. **Example rollouts from ALOHA real-world robot.**

**placing part into a rack**

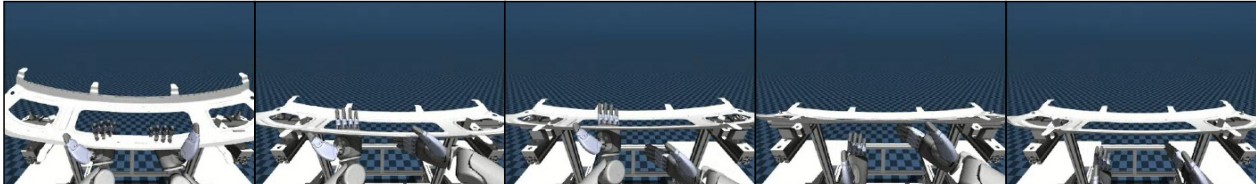

*Figure 7.* **Example episodes from Place Part performed by VinRobotics' simulated VR-H3 humanoid on Mujoco.**

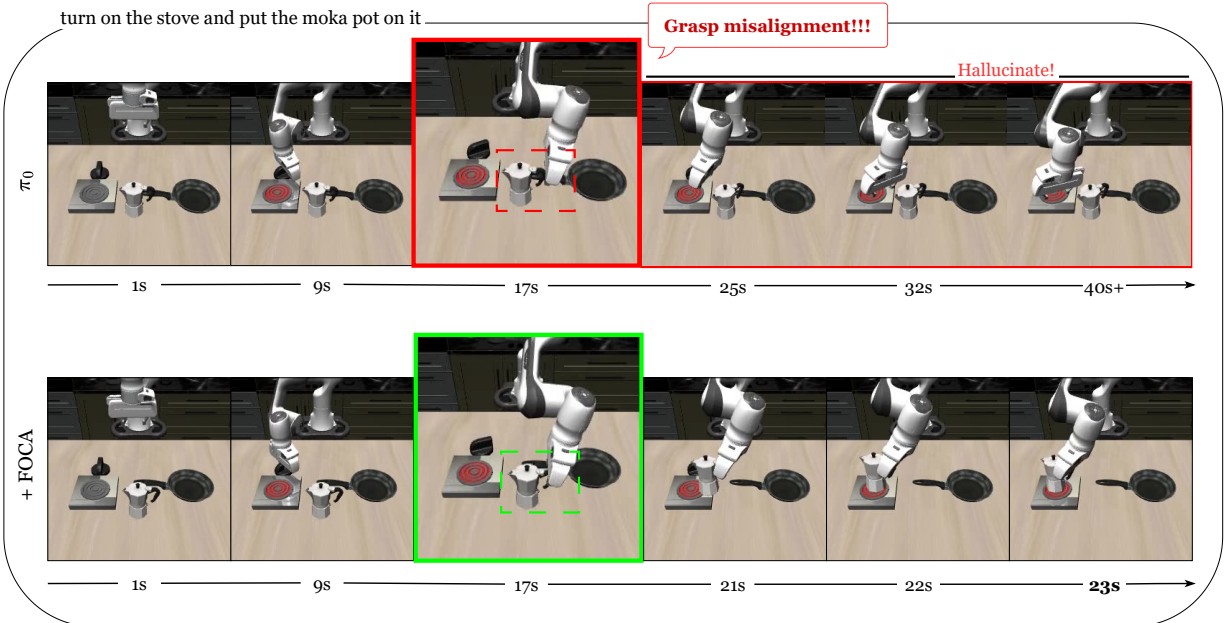

*Figure 8.* **Example rollouts where the baseline $\pi_0$ policy fails (top) and FOCA succeeds (bottom).**

*Table 9.* The number of parameters that FOCA integrates into $\pi_0$ and Gr00tN1.5.

| Method | Total Parameters | Trainable Parameters |
|---|---|---|
| Pi0 | 3.5B | 3.1B |
| Pi0 + FOCA | 3.6B | 3.2B |
| GR00T-N1.5 | 2.4B | 1.1B |
| GR00T-N1.5 + FOCA | 2.5B | 1.2B |

this ground-truth masks combining with task descriptions, we isolate the pixels corresponding to all objects referenced in the task description and gripper then compute the bounding box that tightly encloses every object of interest as shown in Figure 12. By applying this pipeline across all timesteps, we successfully prepare consistent task-grounded spatial region over entire trajectory.

In contrast, for real-world experiments where ground-truth masks are unavailable, we use **Grounding-DINO** (Liu et al., 2024a) to detect all objects specified by the task instruction. The model provides bounding boxes for each queried object class, and we aggregate these detections to output the final object-of-interest areas used at each time steps. This process allows the same task-grounded methodology used in simulation to transfer effectively to real-world settings as shown in Figure 13. Note that these object-of-interest regions are only utilized during training to supervise the Explicit Future Latent Prediction and Implicit Future Alignment modules, and are not required during inference.

**a, Typical hallucinated cases by DreamGen trained with 40% data**

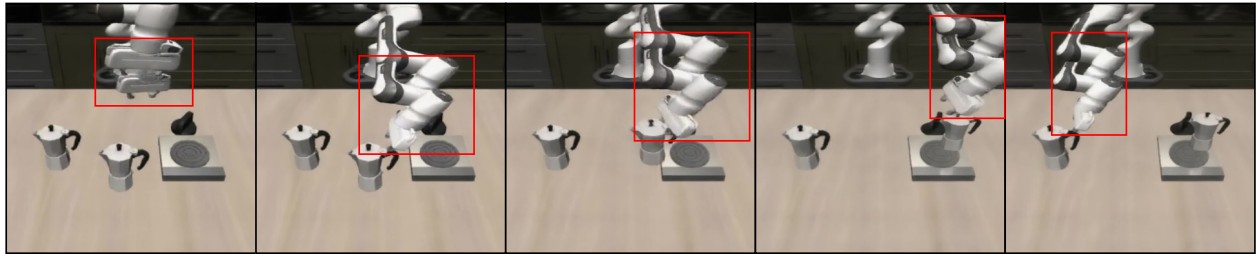

**b, Typical success cases by DreamGen trained with 100% data**

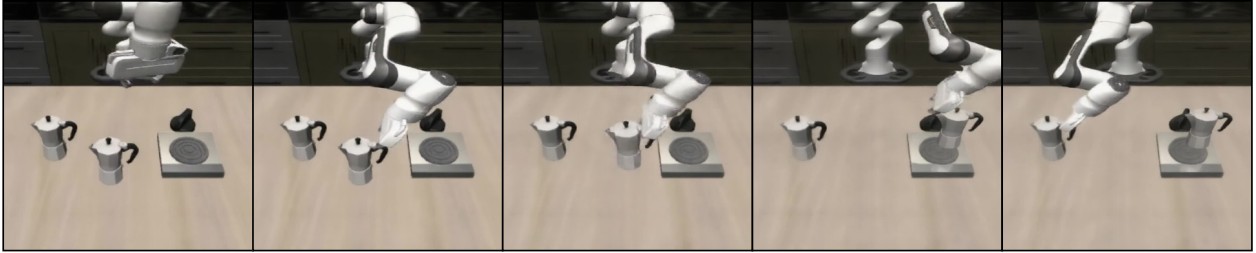

*Figure 9.* **Synthetic rollouts generated by DreamGen at different data scales.** (a) Hallucinated cases with 40% training data; (b) successful cases with 100% training data.

*Table 10.* Our computational overhead when integrating VWM. Importantly, our training structure is consistent with prior pipelines that incorporate video world models. FOCA replaces the pseudo-action generation stage with action-free supervision, rather than introducing an extra stage.

| Training Stages | Training Hours |
|---|---|
| (i) Finetune DreamGen | ~28 hours with 8 H100 (65k steps). This step has a similar cost to the DreamGen paper. |
| (ii) Action-free supervision with FOCA (phase 1) | ~15 hours with 4 A100. DreamGen does not include this step, but extracting pseudo-action from generated videos in (i) takes ~8 hours. |
| (iii) FOCA fine-tuning with explicit and implicit tokens (phase 2) | ~18 hours with 4 A100. This cost is comparable to other baselines. |

### D.3. Computational Resources

For the $\pi_0$ architecture, all experimental settings are trained on four NVIDIA H100 GPUs, while all settings for GR00T-N1.5 are trained on a single NVIDIA H100 GPU. For $\pi_0$, the total training time for the baseline and FOCA on LIBERO is around 18 and 21 hours, respectively, and 20 hours and 23 hours for 100k steps on RoboCasa. For GR00T-N1.5 on RoboCasa, the total training time for the baseline and FOCA is 11 hours and 13 hours for 60k steps, respectively. For detail computational overhead when integrating Video World Model DreamGen, see in Table 10.

### D.4. DreamGen Configurations

In this section, we provide a comprehensive overview of our implementation of the **DreamGen** (Jang et al., 2025) pipeline. This framework is utilized for co-training with synthetic rollouts, video generation, and pseudo-action labeling.

D.4.1. THE FOUR-STAGE WORKFLOW: FROM DATA PREPARATION TO INFERENCE

Our workflow is organized into four distinct stages:

**Step 1: Structured Video Annotation:** Given an original video, we utilize the Qwen 2.5 32B Vision-Language Model (VLM) (Yang et al., 2024) to generate a hierarchical semantic decomposition of the scene. Rather than a simple

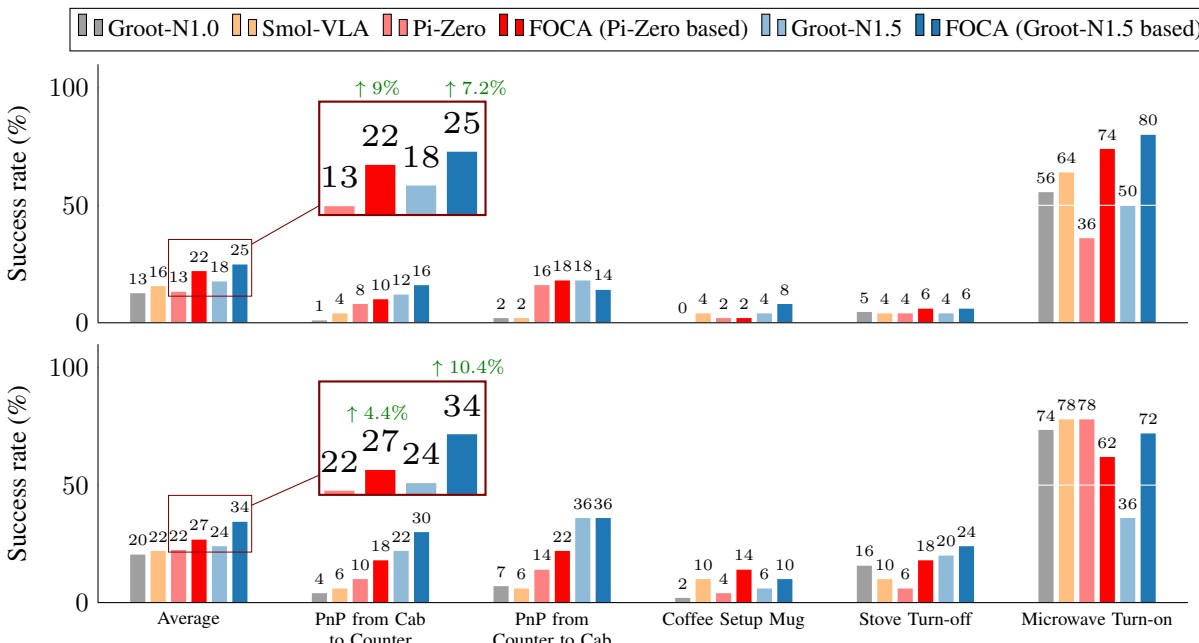

*Figure 10.* FOCA's generalizations performance across Pi-Zero and Groot N1.5 on Robocasa, with 30 demos (top) and 100 demos (bottom) on most 5 challenge tasks

caption, the model produces a structured analysis divided into distinct categories: Setting, Robotic Agent, Task Objectives, Action Sequence, and Environmental Interaction. This structured output acts as a high-fidelity semantic bridge, explicitly isolating the visual features of the robot's morphology and the temporal dynamics of the manipulation task. The standardized system prompt used to enforce this output structure is provided in Figure 14.

**Step 2: World Model Post-Training (Cosmos-Predict 2.0):** We perform domain-specific post-training on the *NVIDIA_Cosmos-Predict2-2B-Video2World* model (Agarwal et al., 2025b). We utilize LIBERO dataset; detailed statistics regarding our data splits are shown in Table 11. The model is trained to predict future video frames conditioned on context frames and a text description. The specific training hyperparameters are detailed in Table 12.

**Step 3: Large-Scale Inference (Synthesis):** Utilizing the fine-tuned checkpoints, we generate synthetic videos with a duration of 17 seconds. The generation is conditioned on an initial screenshot frame from the original video and the corresponding text description generated by the VLM in Step 1.

**Step 4: Pseudo-Action Labeling (IDM):** To extract pseudo-action labels for the generated videos, we train an **Inverse Dynamics Model (IDM)**. The IDM is conditioned on two image frames (start and end) and is trained to predict the action chunks occurring between them using a flow-matching loss objective, as described in the original DreamGen pipeline (Jang et al., 2025). Once trained, the IDM processes the synthetic videos from Step 3 to generate predicted actions (pseudo-actions) for every frame transition.

We adhere to the IDM configuration specified in the DreamGen framework. The model consists of a **SigLIP-2 Vision Encoder** that processes the start and end frames, and an **Action Encoder** that embeds the temporal action sequence. These features are processed by a **Diffusion Transformer** to model the joint distribution. An **Action Decoder** then recovers the final action sequence. We incorporate a feedback loop $k$ between the Action Decoder and Action Encoder to iteratively refine the consistency of the generated pseudo-actions.

D.4.2. STAGE 1: VLM PROMPT ENGINEERING

The fidelity of the synthetic video is highly dependent on the quality of the conditioning text. Figure 14 details the standardized prompt used to extract structured metadata from the original dataset.

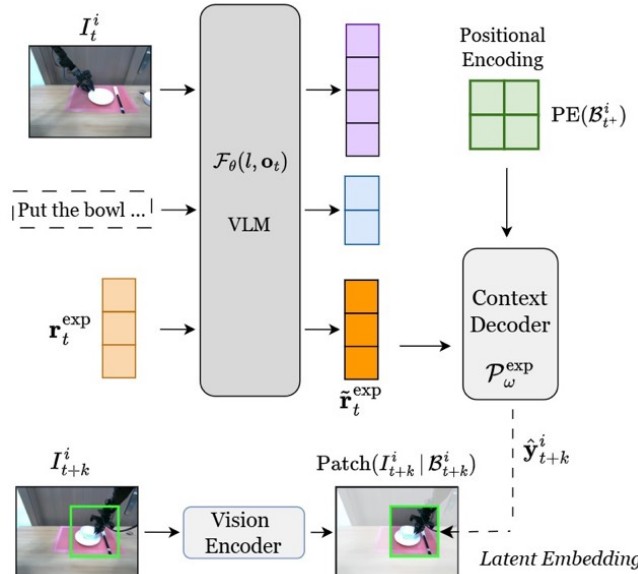

*Figure 11.* Architecture overview of the explicit alignment setting.

*Table 11.* Libero Training Data Splits for DreamGen Implementation.

| Dataset | Length (Frames) | Duration (hr) | FPS | Category |
|---|---|---|---|---|
| Libero (10% Tasks) | 33,587 | 0.46 | 20 | Simulation |
| Libero (40% Tasks) | 133,851 | 1.86 | 20 | Simulation |
| Libero (100% Tasks) | 269,918 | 3.75 | 20 | Simulation |

## E. Simulation Environment and Real-Robot Tasks

In this section, we describe in detail the tasks and datasets used in our experiments, as well as the procedure for generating object-centric bounding boxes for FOCA training. Note that object-centric boxes are generated and used only during training and are not required at inference time.

### E.1. Libero Simulation

**LIBERO** (Liu et al., 2023) dataset consists of 4 subsets: LIBERO-10, LIBERO-Object, LIBERO-Goal, LIBERO-Spatial designed to evaluate different aspects of generalization. Each subset contains 10 tasks, and each task provides 50 demostrations episodes, totaling 500 episodes per subset and 2000 episodes in original dataset. Each episode includes observations from two camera viewpoints: a fixed static `agentview_camera` and an onboard gripper `eye_in_hand_camera`, as illustrated in Figure 15.

To obtain segmentation masks to build Task-grounded Object-centric areas, we re-render the entire dataset using simulator.

*Table 12.* Hyperparameters and Model Configurations for DreamGen Implementation.

| Parameter | Value |
|---|---|
| Base World Model | Cosmos-Predict2-2B-Video2World |
| Resolution | $1280 \times 720$ (720p) |
| Post-Training LR | $1 \times 10^{-4}$ |
| LoRA Config | Rank 4, Alpha 4 |
| RoboCasa and Libero Training | 100 Epochs, Batch Size 32 |

**Task: pick the mug from the counter and place it under the coffee machine dispenser**

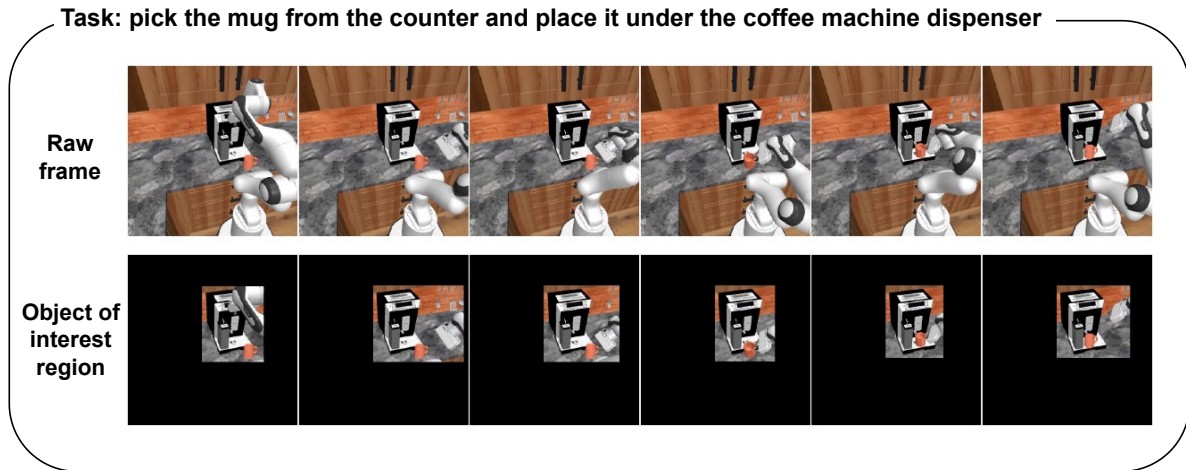

*Figure 12.* **Region of interest in RoboCasa**

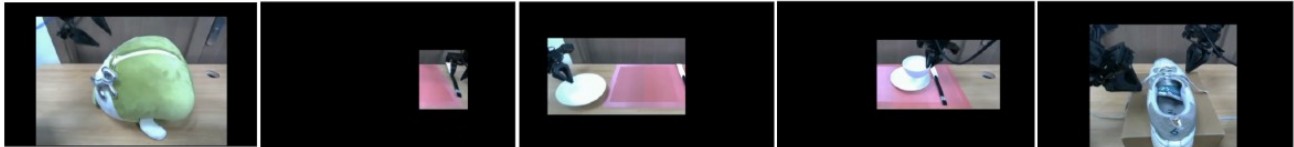

*Figure 13.* **Region of interest in real-world settings**

For each episode, we reset the enviroment to its saved initial state and replay each action from the demotration labels. During re-rendering, we record observations that now include segmentation masks, which are not provided in the original datasets. Moreover, to ensure the constructed dataset maintains high quality, we perform additional filtering to remove no-op actions, since the produce no change in the robot state as well as discard unsuccessful demostrations, which could introduce inconsistent or misleading supervision. After this process, the final dataset reduces from 2000 episodes to 1676 valid demostrations. For experiments with different scale datasets, we construct 10% and 40% subsets by random sampling 5 episodes and 20 episodes per tasks for each subset respectively. After training, we evaluate models using 50 trials per task, and the results are summarized in Figure 2.

### E.2. Robocasa Simulation

**RoboCasa Kitchen** (Nasiriany et al., 2024) is a large-scale simulation benchmark designed to model everyday manipulation tasks with different kitchen environments. It includes 24 tasks, each providing apprriximately 3000 demostrations, along with numerous predefined subsets. Each demonstration provides three synchronized visual viewpoints from 3 different cameras include: `agentview_left_camera`, `agentview_right_camera`, `eye_in_hand_camera` as shown in Figure 15. In our experiments, we adopt the 30-demos and 100-demos configuration. Besides, the tasks in RoboCasa can be categorize into different types which are pick-and-place, turn on/off, open/close, and others. From the full set of 24 tasks, we select 5 representative tasks for our experiments include `PnPCabToCounter`, `PnPCounterToCab`, `CoffeeSetupMug`, `TurnOffStove`, `TurnOnMicrowave`. These tasks cover a range of obkect interaction and scene configurations, making them suitable for evaluating task-grounded object-centric modeling.

To contruct the consistent object of interest regions accross all timesteps for selected RoboCasa demostrations, we apply the same re-rendering and segmentation-mask extraction process used for LIBERO mentioned above. We perform 50 evaluation trials for each selected task and report the outcomes in Figure 3c.

**System Prompt for Structured Video Annotation**

**System Instruction:**
You are an expert in robotic manipulation analysis. Watch the video and generate a structured description. Strictly follow this output format:

**1. Setting:**
  • Describe the environment type (e.g., kitchen, lab).
  • List key features (furniture, lighting, surfaces).

**2. Robotic Arm:**
  • Describe the robot's appearance (color, joints) and design style.
  • Note its position relative to target objects/containers.

**3. Task:**
  • Define the specific operation (e.g., reaching, retrieving).
  • Identify the target object and its location.

**4. Action Sequence:**
  • Detail the maneuvering of joints and end-effector interaction.
  • Describe the smoothness and precision of the movement.

**5. Environment Interaction:**
  • Analyze how the robot navigates constraints (e.g., open doors, confined spaces).

**6. Focus on Details:**
  • Emphasize fine motor control and specific design elements.

**7. Purpose:**
  • Summarize the broader capability demonstrated (e.g., home automation).

**Assistant's Generated Output (Example Segment):**

**1. Setting:** The video is set in a kitchen environment...
**2. Robotic Arm:** The robotic arm is white with black joints...
**3. Task:** The arm extends its joints to reach inside the cabinet... ...

*Figure 14.* Standardized prompt template for Stage 1: The VLM is instructed to decompose the video into a 7-point structured attribute analysis.

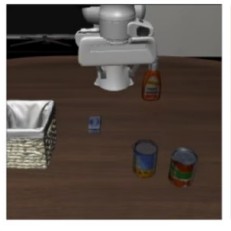 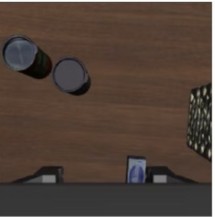 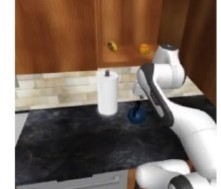 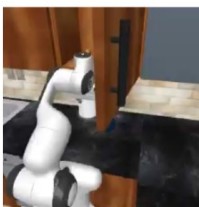 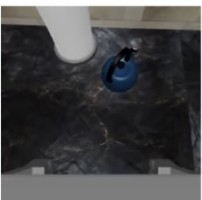

*(a)* LIBERO camera views          *(b)* RoboCasa camera views

*Figure 15.* **Camera views in LIBERO and RoboCasa**

### E.3. ALOHA Tasks

We evaluate our method on three real-robot manipulation tasks using the Mobile ALOHA platform. All tasks require bimanual manipulation and multi-view visual observations. The experiments are executed on a single workstation equipped with an NVIDIA RTX 5090 GPU. Task definitions, initialization, and the number of demonstrations are described below.

**Open Bag.** The robot holds the bag with its left arm and grasps the zipper keychain with its right arm. The task is completed successfully when the bag is fully opened. The bag is initialized near the center of the table with a uniformly random planar position offset of 0 to 5 cm and a random in-plane rotation of 0 to 10°. Each demonstration consists of 240 control steps (16 seconds). We collect 100 demonstrations.

**Tie Shoelaces.** The robot grasps the two shoelaces with both arms, forms a loop, and tightens the knot by pulling the laces in opposite directions. The task is completed when the knot is tightened. The shoe is initialized near the center of the workspace with a uniformly random planar position offset of 0 to 5 cm and a random in-plane rotation of 0 to 10°. Each demonstration consists of 300 control steps (20 seconds). We collect 300 demonstrations.

**Set Table.** The robot places chopsticks onto the table mat using the right arm, places a plate at the center of the mat using the left arm, and places a bowl at the center of the plate using the right arm. The table mat is initialized with a uniformly random horizontal offset of 0 to 5 cm. The chopsticks and bowl are initialized randomly on the left-arm side of the workspace, and the plate is initialized randomly on the right-arm side, subject to reachability constraints. Each demonstration consists of 600 control steps (40 seconds). We collect 300 demonstrations.

### E.4. UR5 Tasks

We further evaluate our method on an additional robotic embodiment, the UR5. We employ two different variations UR5 arms, one UR5e equipped with a Robotiq Hand-E gripper, and one UR5 is equipped with a OnRobot-RG2 gripper. Two camera views are provided: one mounted on the table and the other on the robot wrist. On this real-robot platform, we design three handcrafted tabletop tasks, as illustrated in Figure 16 and Figure 17.

**Dispense Napkin**: The robot dispenses a napkin from a container. The napkin box is uniformly distributed between these environments, where the napkin box is moved around within an area of 50x30 cm and rotated between 0-90 degrees. The task is considered successful if the napkin is fully dispensed. The trajectories for this task are on average about 95 frames long, we evaluate each trial with a maximum limit of 60 seconds: if the task has not been completed by then, the episode is counted as a failure.

**Handle Bulk Material**: The robot picks up a bowl containing bulk material and pours the material into a designated target container.

**Short Chemistry Test Tube**: The robot picks up the chemistry test tube with the prompted content color and place into a designated target.

For each task, we collect 100 demonstrations for training (correspond to 100% training data regimes). The experiments are executed on a single workstation equipped with an NVIDIA RTX 5090 GPU.

**Results** Figure 18 highlights the effectiveness of FOCA in real-world robotic manipulation tasks on the UR5 platform. Across all three evaluated tasks, FOCA consistently outperforms the baseline VLA model under both partial-data (40%) and full-data (100%) training settings. The improvement is especially noticeable in the more challenging manipulation scenarios:

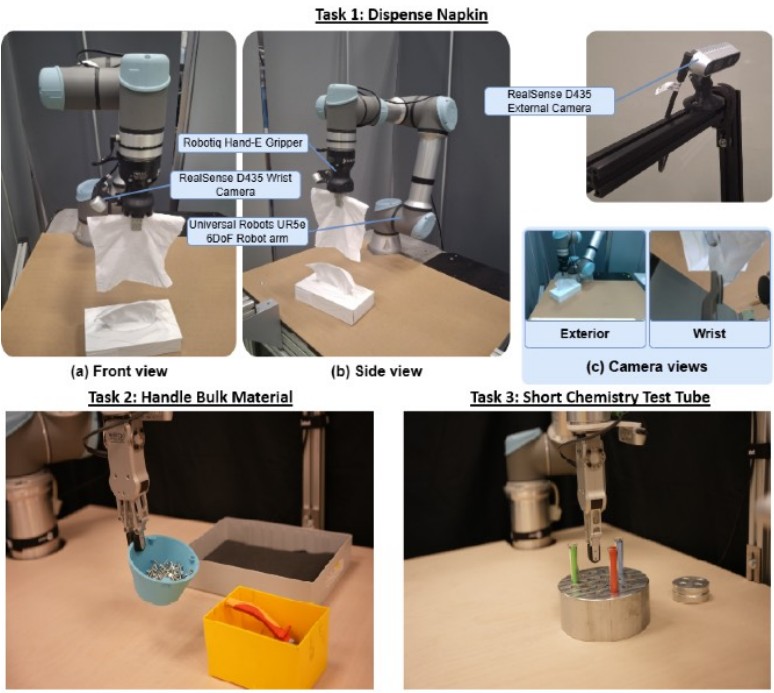

*Figure 16.* The three handcrafted tabletop tasks with UR5.

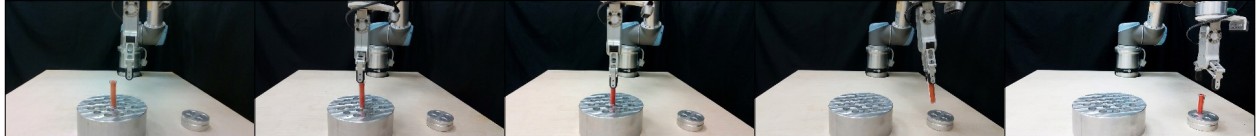

*Figure 17.* Example of FOCA rollout episode from UR5.

- Handle bulk material (UR5): FOCA achieves the largest gain, improving success rates from 30% → 50% under full-data training, while the baseline completely fails under the 40% setting (0%). This suggests that FOCA learns more robust action representations and generalizes better to complex contact-rich interactions.

- Dispense napkin (UR5e): FOCA improves performance from 20% → 45% using 100% data and from 10% → 25% with only 40% data. The consistent gap indicates stronger policy reliability even when training data is limited.

- Short chemistry test tube (UR5): FOCA again surpasses the baseline in both settings, reaching 25% success compared to 15% for Base-VLA under full-data training. Although the absolute scores are lower due to the precision required in this task, FOCA still demonstrates better fine-grained manipulation capability.

All in all, these results show two key strengths of FOCA: (1) higher task success rates across diverse real-world manipulation tasks; and (2) better data efficiency, since FOCA maintains clear advantages even when trained with only 40% of the dataset. This indicates that FOCA improves both the robustness and generalization ability of the manipulation policy, particularly in difficult real-world settings where perception noise, contact dynamics, and limited training data are major challenges.

### E.5. Place Part

We implement this task in MuJoCo (Todorov et al., 2012) using a humanoid robot. Figure 19a illustrates our multi-view camera setup, providing observations to the policy. This task evaluates the robot's ability to perceive, align, and insert parts under tight constraints. We record synchronized observations from three RGB cameras that provide complementary viewpoints, using a teleoperation method in simulation.

Robot proprioception is represented by a 26-dimensional joint-state vector, comprising:

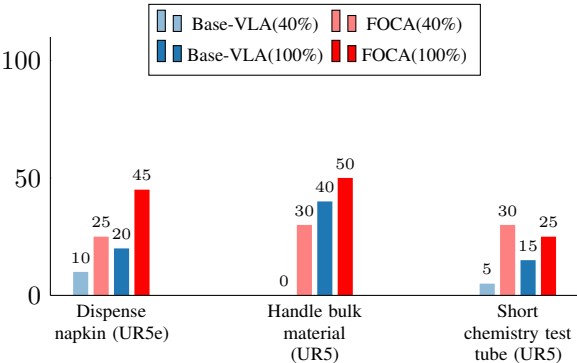

*Figure 18.* Real-world results of the three manipulation tasks on UR5

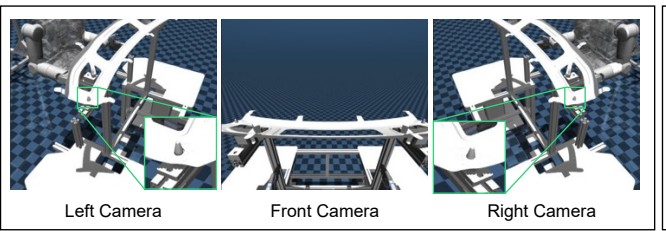

(a) Camera views

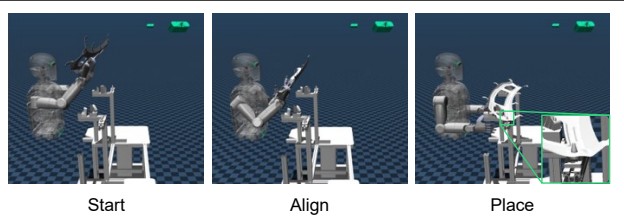

(b) Task execution stages

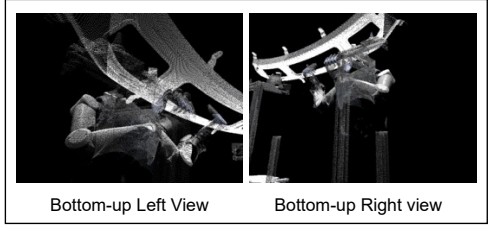

(c) Virtual views

*Figure 19.* **Place Part in MuJoCo: camera views, key execution stages, and virtual views**. (a) Left, front, and right camera views. (b) Task stages: Start, Align, and Place. (c) Bottom-up virtual views. Green annotations illustrate the strict spatial alignment constraints of the task, requiring the two holes on either side of the part to be precisely aligned with and seated onto the two small pins on either side of the jig.

```
1   left_shoulder_pitch_joint      14 right_wrist_pitch_joint
2   left_shoulder_roll_joint       15 L_thumb_swing_joint
3   left_shoulder_yaw_joint        16 L_thumb_1_joint
4   left_elbow_pitch_joint         17 L_index_1_joint
5   left_wrist_yaw_joint           18 L_middle_1_joint
6   left_wrist_roll_joint          19 L_ring_1_joint
7   left_wrist_pitch_joint         20 L_pinky_1_joint
8   right_shoulder_pitch_joint     21 R_thumb_swing_joint
9   right_shoulder_roll_joint      22 R_thumb_1_joint
10  right_shoulder_yaw_joint       23 R_index_1_joint
11  right_elbow_pitch_joint        24 R_middle_1_joint
12  right_wrist_yaw_joint          25 R_ring_1_joint
13  right_wrist_roll_joint         26 R_pinky_1_joint
```

Figure 19b illustrates the key stages of a successful sequence. At first, the robot is initialized in a raised, overhead grasping pose while holding the target part. The model is then started to control the robot to adjust the part's position and orientation relative to the jig. Successful completion is achieved when the part is placed and stably seated in the jig, with the two holes

on either side of the part precisely aligned with and seated onto the two small pins on either side of the jig.

Additionally, we use a render in RVT-2 (Goyal et al., 2024) to generate two virtual views (Figure 19c) in order to reduce occlusions and provide more information. Specifically, we reconstruct point clouds from RGB-D images and the camera intrinsics, and transform them into a shared world coordinate frame using the camera extrinsics. Virtual RGB-D images are then rendered by projecting these points onto two 2D image planes.

## F. Theoretical Investigation

### F.1. Problem setting and notation

We model the environment as a discounted MDP $\mathcal{M} = (\mathcal{S}, \mathcal{A}, P, \gamma)$ with $\gamma \in (0, 1)$. A *state* $s \in \mathcal{S}$ represents the VLA input at a time step (e.g., image observation, robot state, and instruction), and $a \in \mathcal{A}$ is an action. We are given an offline dataset of expert trajectories $\mathcal{D} = \{\tau\}$, where $\tau = (s_0, a_0, s_1, a_1, \ldots, s_T)$ is generated by a behavior policy $\mu(a \mid s)$. We write expectations over $\mathcal{D}$ as $\mathbb{E}_{\tau \sim \mathcal{D}}[\cdot]$.

**Mapping to main-text notation.** In the main paper, the context for implicit future alignment is the projected implicit token $x_t := z_t$ (Eq. (8)), which is a deterministic function of the VLA input $s_t$. The goal variable is the future interaction embedding $g_{t+k} := y_{t+k} = h(s_{t+k})$ (Eq. (9)). Negatives are drawn from a proposal $p_{\text{neg}}$, instantiated in FOCA by the task-mismatched pool $\mathcal{N}_{\neg\text{task}}$ (Eq. (10)). For readability, the analysis below is written in terms of $s_t$ (or $(s_t, a_t)$); the same results apply when conditioning on $x_t$ by treating the representation as part of the observation. We use $\mathcal{L}_{\text{fm}}$ to denote the action-supervised loss (flow matching for diffusion policies) and $\mathcal{L}_{\text{imp}}$ for the implicit InfoNCE loss.

**Measure-theoretic note.** For clarity, proofs are written for the *discrete* case where probabilities are well-defined. For continuous/high-dimensional observations, interpret $\rho^\pi(\cdot \mid x)$ and $p_{\text{neg}}(\cdot)$ as densities w.r.t. a common base measure (or as measures with Radon–Nikodym derivatives). In particular, statements involving $\log \rho^\pi(g \mid x) - \log p_{\text{neg}}(g)$ require absolute continuity of $\rho^\pi(\cdot \mid x)$ w.r.t. $p_{\text{neg}}$ on the support of interest.

**Object-centric structure.** Let $\xi(s)$ be an object extractor returning $m$ object regions (e.g., masks/boxes): $\xi(s) = \{r^{(i)}(s)\}_{i=1}^m$. We consider an encoder producing a global embedding and object embeddings:

$$\psi(s) = \left(\psi^{\text{g}}(s), \psi^{(1)}(s), \ldots, \psi^{(m)}(s)\right) \in \mathbb{R}^{d \times (1+m)}. \tag{17}$$

We do not assume $\xi$ is perfect; its errors will enter as a misspecification term.

**Goal space.** Let $\mathcal{G}$ denote a goal space and let $h : \mathcal{S} \to \mathcal{G}$ be a measurable *goal map*. We write $g_t := h(s_t)$. In our setting, $h$ may be the identity map id (full-state goals) or an object-centric extractor (e.g., a cropped region/mask, or a learned token derived from $r^{(i)}(s_t)$). When $h = \text{id}$ we have $\mathcal{G} = \mathcal{S}$ and we may write $g_t = s_t$ for convenience.

### F.2. Discounted future occupancy and Bellman recursions

We define the *discounted future-occupancy (reachability)* distribution under policy $\pi$ (Dayan, 1993; Schaul et al., 2015):

**Definition F.1** (Discounted future occupancy). Fix a policy $\pi$ and write $g_t := h(s_t) \in \mathcal{G}$. For a state $s$, define the *state-conditional* discounted future-occupancy (reachability) distribution:

$$\rho^\pi(g \mid s) := (1 - \gamma) \sum_{k=1}^\infty \gamma^{k-1} \Pr_\pi(g_{t+k} = g \mid s_t = s), \tag{18}$$

and, if we additionally condition on the initial action, define the *action-conditional* occupancy:

$$\rho^\pi(g \mid s, a) := (1 - \gamma) \sum_{k=1}^\infty \gamma^{k-1} \Pr_\pi(g_{t+k} = g \mid s_t = s, a_t = a). \tag{19}$$

**Proposition F.2** (Occupancy Bellman equations (Dayan, 1993; Schaul et al., 2015)). *Fix a policy $\pi$ and a goal $g \in \mathcal{G}$. Define the state value $V_g^\pi(s) := \rho^\pi(g \mid s)$ and the action value $Q_g^\pi(s, a) := \rho^\pi(g \mid s, a)$. Let $P^\pi(s' \mid s) := \mathbb{E}_{a \sim \pi(\cdot \mid s)}[P(s' \mid s, a)]$*

*denote the policy-induced state transition kernel. Then the state-conditional occupancy satisfies*

$$V_g^\pi(s) = (1 - \gamma) \Pr_\pi(g_{t+1} = g \mid s_t = s) +$$
$$\gamma \, \mathbb{E}_{s' \sim P^\pi(\cdot|s)} \big[ V_g^\pi(s') \big], \tag{20}$$

*and the action-conditional occupancy satisfies*

$$Q_g^\pi(s, a) = (1 - \gamma) \Pr(g_{t+1} = g \mid s_t = s, a_t = a) +$$
$$\gamma \, \mathbb{E}_{s' \sim P(\cdot|s,a), \, a' \sim \pi(\cdot|s')} \big[ Q_g^\pi(s', a') \big]. \tag{21}$$

*Equivalently, $V_g^\pi$ and $Q_g^\pi$ are the state-/action-value functions for the sparse goal-reaching reward $r_g(s, a, s') = (1 - \gamma) \mathbf{1}\{h(s') = g\}$.*

*Proof.* We first prove the action-conditional recursion (21) for $Q_g^\pi(s, a) := \rho^\pi(g \mid s, a)$.

By Definition F.1,

$$Q_g^\pi(s, a) = \rho^\pi(g \mid s, a) = (1 - \gamma) \sum_{k=1}^{\infty} \gamma^{k-1} \Pr_\pi(g_{t+k} = g \mid s_t = s, a_t = a).$$

Separate the $k = 1$ term:

$$Q_g^\pi(s, a) = (1 - \gamma) \Pr_\pi(g_{t+1} = g \mid s_t = s, a_t = a) + (1 - \gamma) \sum_{k=2}^{\infty} \gamma^{k-1} \Pr_\pi(g_{t+k} = g \mid s_t = s, a_t = a).$$

For $k \geq 2$, apply the law of total expectation over $(s_{t+1}, a_{t+1})$:

$$\Pr_\pi(g_{t+k} = g \mid s_t = s, a_t = a) = \mathbb{E}_{s' \sim P(\cdot|s,a), \, a' \sim \pi(\cdot|s')} \Big[ \Pr_\pi(g_{t+k} = g \mid s_{t+1} = s', a_{t+1} = a') \Big].$$

Let $j = k - 1 \geq 1$. Then $g_{t+k} = g_{t+1+j}$. Hence

$$(1 - \gamma) \sum_{k=2}^{\infty} \gamma^{k-1} \Pr_\pi(g_{t+k} = g \mid s_t = s, a_t = a) = (1 - \gamma) \sum_{j=1}^{\infty} \gamma^j \, \mathbb{E}_{s',a'} \Big[ \Pr_\pi(g_{t+1+j} = g \mid s_{t+1} = s', a_{t+1} = a') \Big]$$

$$= \gamma \, \mathbb{E}_{s',a'} \Big[ (1 - \gamma) \sum_{j=1}^{\infty} \gamma^{j-1} \Pr_\pi(g_{t+1+j} = g \mid s_{t+1} = s', a_{t+1} = a') \Big]$$

$$= \gamma \, \mathbb{E}_{s',a'} \big[ \rho^\pi(g \mid s', a') \big] = \gamma \, \mathbb{E}_{s',a'} \big[ Q_g^\pi(s', a') \big],$$

where $s' \sim P(\cdot \mid s, a)$ and $a' \sim \pi(\cdot \mid s')$. Combining terms yields Eq. (21).

For the state-conditional recursion (20), note that by the law of total probability, $V_g^\pi(s) = \rho^\pi(g \mid s) = \mathbb{E}_{a \sim \pi(\cdot|s)}[\rho^\pi(g \mid s, a)] = \mathbb{E}_{a \sim \pi(\cdot|s)}[Q_g^\pi(s, a)]$. Taking expectation of (21) over $a \sim \pi(\cdot \mid s)$ and using $P^\pi(s' \mid s) = \mathbb{E}_{a \sim \pi(\cdot|s)}[P(s' \mid s, a)]$ yields Eq. (20). $\square$

*Remark* F.3 (What is actually learned?). FOCA's implicit future alignment operates on the model's *representations* of the current observation/instruction, and does *not* condition on action outputs. Therefore, the implicit objective targets the state-conditional occupancy/value $V_g^\mu(s) := \rho^\mu(g \mid s)$ under the demonstration distribution. If one augments the context to include an action embedding (an optional extension), the same analysis would instead target the action-conditional occupancy $Q_g^\mu(s, a) := \rho^\mu(g \mid s, a)$. Both satisfy Bellman-style recursions (Proposition F.2), but only the latter has a direct $Q$-function interpretation.

### F.3. Implicit alignment as density-ratio estimation (InfoNCE $\Rightarrow$ occupancy/value)

We formalize the implicit module as learning a score $f_\theta$ between a *context* (current state or state-action) and a *goal* (future state), trained with negatives from a proposal distribution $p_{\text{neg}}$.

**Positive sampling scheme.** A critical detail is how we sample future positives. Let $k \sim \text{Geom}(1 - \gamma)$ over $\{1, 2, \dots\}$, i.e., $\Pr(k) = (1 - \gamma)\gamma^{k-1}$, and let $g = g_{t+k} = h(s_{t+k})$. If we take the context to be the current state $x = s_t$ (as in FOCA's implicit alignment), then the induced conditional distribution of $g$ is exactly $\rho^\mu(\cdot \mid s_t)$. (For completeness, if the context includes the initial action $x = (s_t, a_t)$, the same sampling yields $\rho^\mu(\cdot \mid s_t, a_t)$.)

**Lemma F.4** (Geometric sampling matches discounted occupancy). *If $k \sim \mathrm{Geom}(1 - \gamma)$ and $g = g_{t+k}$, then $\Pr(g = \tilde{g} \mid x_t) = \rho^\mu(\tilde{g} \mid x_t)$ for all $\tilde{g} \in \mathcal{G}$.*

*Proof.* Fix $(s_t, a_t) = (s, a)$. Let $k \sim \mathrm{Geom}(1 - \gamma)$ on $\{1, 2, \dots\}$ so that $\Pr(k) = (1 - \gamma)\gamma^{k-1}$, and let $g = g_{t+k}$. Then for any $\tilde{g} \in \mathcal{G}$,

$$\Pr(g = \tilde{g} \mid s, a) = \sum_{k=1}^{\infty} \Pr(k) \Pr_\mu(g_{t+k} = \tilde{g} \mid s_t = s, a_t = a)$$

$$= (1 - \gamma) \sum_{k=1}^{\infty} \gamma^{k-1} \Pr_\mu(g_{t+k} = \tilde{g} \mid s_t = s, a_t = a) = \rho^\mu(\tilde{g} \mid s, a),$$

where the last equality is Definition F.1 with $\pi = \mu$. $\qquad\square$

Since FOCA uses state-only context $x = s_t$, the same sampling scheme yields $p^+(g \mid s) = \rho^\mu(g \mid s)$. (If one conditions on $(s, a)$ instead, Lemma F.4 gives $p^+(g \mid s, a) = \rho^\mu(g \mid s, a)$.)

**Contrastive objective.** Let $x$ denote the context (either $(s, a)$ or $s$). Let $f_\theta(x, g) \in \mathbb{R}$ be a score; in practice this is implemented by dot-product similarity between learned embeddings. We sample a positive $g^+ \sim p^+(g \mid x)$ and $N$ negatives $g_1^-, \dots, g_N^-$ i.i.d. $p_{\mathrm{neg}}$ (Eysenbach et al., 2022; Zheng et al., 2023). We optimize the InfoNCE loss (Gutmann & Hyvärinen, 2012; Oord et al., 2018):

$$\mathcal{L}_{\mathrm{imp}}(\theta) = -\mathbb{E}\left[ \log \frac{\exp(f_\theta(x, g^+))}{\exp(f_\theta(x, g^+)) + \sum_{j=1}^{N} \exp(f_\theta(x, g_j^-))} \right]. \tag{22}$$

**Lemma F.5** (Bayes-optimal logit is a density ratio). *Consider binary classification between samples from $p^+(g \mid x)$ (positive) and $p_{\mathrm{neg}}(g)$ (negative). The Bayes-optimal logit $f^*(x, g)$ satisfies*

$$f^*(x, g) = \log p^+(g \mid x) - \log p_{\mathrm{neg}}(g) + c(x) \tag{23}$$

*for an arbitrary function $c$ that does not depend on $g$.*

*Proof.* Consider binary labels $Y \in \{+, -\}$ with conditional distributions $G \mid (X = x, Y = +) \sim p^+(\cdot \mid x)$ and $G \mid (Y = -) \sim p_{\mathrm{neg}}(\cdot)$. Let $\pi_+(x) = \Pr(Y = + \mid X = x)$ and $\pi_-(x) = 1 - \pi_+(x)$. Bayes' rule gives

$$\Pr(Y = + \mid x, g) = \frac{\pi_+(x) p^+(g \mid x)}{\pi_+(x) p^+(g \mid x) + \pi_-(x) p_{\mathrm{neg}}(g)}.$$

Hence the Bayes-optimal log-odds is

$$\log \frac{\Pr(Y = + \mid x, g)}{\Pr(Y = - \mid x, g)} = \log \frac{\pi_+(x) p^+(g \mid x)}{\pi_-(x) p_{\mathrm{neg}}(g)} = \log p^+(g \mid x) - \log p_{\mathrm{neg}}(g) + \log \frac{\pi_+(x)}{\pi_-(x)}.$$

Setting $c(x) := \log \frac{\pi_+(x)}{\pi_-(x)}$ yields the claim. $\qquad\square$

*Proof of Theorem 4.1.* We use a standard "one-positive among negatives" generative model for InfoNCE. Fix a context $x$. Sample an index $I$ uniformly from $\{0, 1, \dots, N\}$. Sample $g_I \sim p^+(\cdot \mid x)$ and $g_j$ i.i.d. $p_{\mathrm{neg}}(\cdot)$ for $j \neq I$. Let $\mathcal{G} = \{g_0, \dots, g_N\}$ denote the resulting multiset.

The InfoNCE loss in (22) is the expected multiclass cross-entropy for predicting $I$ from $\mathcal{G}$ using the softmax model

$$q_\theta(I = i \mid x, \mathcal{G}) = \frac{\exp(f_\theta(x, g_i))}{\sum_{j=0}^{N} \exp(f_\theta(x, g_j))}.$$

For any fixed $(x, \mathcal{G})$, cross-entropy is uniquely minimized (a.s.) when the model matches the true posterior: $q_\theta(\cdot \mid x, \mathcal{G}) = \Pr(\cdot \mid x, \mathcal{G})$.

We therefore compute $\Pr(I = i \mid x, \mathcal{G})$. By Bayes' rule and conditional independence,

$$\Pr(\mathcal{G} \mid x, I = i) = p^+(g_i \mid x) \prod_{j \neq i} p_{\mathrm{neg}}(g_j) = \left( \frac{p^+(g_i \mid x)}{p_{\mathrm{neg}}(g_i)} \right) \prod_{j=0}^{N} p_{\mathrm{neg}}(g_j).$$

Since the factor $\prod_{j=0}^{N} p_{\mathrm{neg}}(g_j)$ does not depend on $i$ and the prior on $I$ is uniform, we obtain

$$\Pr(I = i \mid x, \mathcal{G}) \propto \frac{p^+(g_i \mid x)}{p_{\mathrm{neg}}(g_i)} \implies \Pr(I = i \mid x, \mathcal{G}) = \frac{\exp(\log p^+(g_i \mid x) - \log p_{\mathrm{neg}}(g_i))}{\sum_{j=0}^{N} \exp(\log p^+(g_j \mid x) - \log p_{\mathrm{neg}}(g_j))}.$$

Thus a sufficient condition for $q_\theta(\cdot \mid x, \mathcal{G})$ to match the posterior for all sets $\mathcal{G}$ is

$$f_\theta(x, g) = \log p^+(g \mid x) - \log p_{\text{neg}}(g) + c(x),$$

where $c(x)$ is any function independent of $g$ (it cancels in the softmax). This proves the density-ratio form.

Finally, by Lemma F.4, geometric future sampling implies $p^+(g \mid x) = \rho^\mu(g \mid x)$, yielding (15). $\qquad \square$

*Remark* F.6 (Optional connection to (soft) $Q$-learning and RL-as-inference). Equation (15) implies that $\exp(f_\theta)$ estimates an *unnormalized* goal-conditioned occupancy (Ziebart et al., 2008). This already yields a value-like reachability score in the state-conditioned setting used by FOCA.

If one instead uses an *action-conditioned* score $f_\theta(s, a, g)$ (an optional extension, not required by FOCA's implicit alignment), one may interpret $f_\theta$ as an energy over actions for a fixed goal $g$ and define an energy-based policy $\pi_\theta(a \mid s, g) \propto \exp(f_\theta(s, a, g))$, which is structurally aligned with maximum-entropy RL. Making this connection rigorous in high-dimensional action spaces requires additional assumptions and a gauge-fixing/normalization to remove additive terms that depend on $(s, a)$; otherwise the InfoNCE optimum is identifiable only up to $c(s, a)$.

*Remark* F.7 (Critical caveat: negative sampling matters). If $p_{\text{neg}}$ is far from the marginal goal distribution induced by the dataset, the ratio in (15) may prioritize "easy negatives" and distort the learned metric (Zheng et al., 2023). Empirically, this manifests as unstable or task-irrelevant alignment unless negatives are carefully chosen. In FOCA we draw negatives from the task-mismatched pool $\mathcal{N}_{\neg\text{task}}$ (Eq. (10)), which can be viewed as choosing a task-conditioned proposal that yields informative (hard) negatives.

## F.4. Multi-frame futures and $n$-step structure

Theorems above hold exactly for geometric sampling. If instead one chooses a fixed horizon $k$, then $p^+(g \mid x)$ becomes the $k$-step transition distribution, not the discounted occupancy. This is still useful, but it no longer corresponds to a Bellman fixed point for a single discount.

**Proposition F.8** (Fixed-horizon positives learn $k$-step transitions). *If positives are sampled as $g = g_{t+k} = h(s_{t+k})$ with deterministic $k \geq 1$, then $f_\theta(x, g)$ estimates $\log \Pr_\mu(g_{t+k} = g \mid x) - \log p_{\text{neg}}(g)$ up to $c(x)$.*

*Proof.* Under fixed-horizon sampling $g = g_{t+k}$ (deterministic $k \geq 1$), the positive conditional distribution is

$$p^+(g \mid x) = \Pr_\mu(g_{t+k} = g \mid x),$$

where $x$ denotes the chosen context (state or state-action). Applying the argument in the proof of Theorem 4.1 with this $p^+$ gives

$$f_\theta(x, g) = \log \Pr_\mu(g_{t+k} = g \mid x) - \log p_{\text{neg}}(g) + c(x),$$

as claimed. In the special case $h = \text{id}$, then $g_{t+k} = s_{t+k}$ and the positive distribution reduces to the standard $k$-step state transition. $\qquad \square$

*Remark* F.9 (Practical implication). Using a mixture over multiple horizons corresponds to learning a mixture of $k$-step transition distributions (Oord et al., 2018). Choosing a geometric distribution is the unique memoryless choice (up to constants) that yields a single discounted occupancy target with a Bellman-style fixed point (Proposition F.2).

## F.5. Explicit object-centric prediction as restricted world modeling

FOCA's explicit head predicts a future interaction embedding in latent space (Sec. 3.2), focusing supervision on task-grounded dynamic regions. Abstractly, let $x$ denote the current context (in FOCA, $x = x_t = z_t$) and let $g^+ = h(s_{t+k})$ denote the sampled future interaction embedding. The explicit loss in the main paper (Eq. (6)) is a deterministic regression objective:

$$\mathcal{L}_{\text{exp}}(\theta) = \mathbb{E}\big[\|\widehat{g}_\theta(x) - g^+\|_2^2\big], \qquad g^+ \sim p^+(g \mid x). \tag{24}$$

*Proof of Proposition 4.2.* Fix a context $x$. The conditional objective is $\mathcal{L}_{\text{exp}}(x; \widehat{g}) = \mathbb{E}[\|\widehat{g}(x) - g^+\|_2^2 \mid x]$. Expanding the square and dropping terms independent of $\widehat{g}(x)$ gives $\mathcal{L}_{\text{exp}}(x; \widehat{g}) = \|\widehat{g}(x)\|_2^2 - 2\widehat{g}(x)^\top \mathbb{E}[g^+ \mid x] + \text{const}$. This quadratic is minimized uniquely by $\widehat{g}^*(x) = \mathbb{E}[g^+ \mid x]$. Under geometric future sampling (Lemma F.4), $g^+ \mid x \sim \rho^\mu(\cdot \mid x)$, hence $\widehat{g}^*(x) = \mathbb{E}_{g \sim \rho^\mu(\cdot \mid x)}[g]$. $\qquad \square$

*Remark* F.10 (Connection to likelihood modeling). The squared loss (24) is equivalent to a Gaussian negative log-likelihood with fixed variance, so Proposition 4.2 can be interpreted as maximum-likelihood estimation of the Gaussian mean. However, unlike the implicit contrastive objective (Theorem 4.1), this regression loss does not estimate the full occupancy distribution in multimodal futures; it summarizes it through a point estimate, which is why it complements (rather than replaces) implicit alignment in practice.

## F.6. Hybrid objective and object-centric factorization

We combine action supervision (flow matching / imitation) with future-oriented auxiliary losses:

$$\mathcal{L}(\theta) = \mathcal{L}_{\text{fm}}(\theta) + \lambda_{\text{imp}}\mathcal{L}_{\text{imp}}(\theta) + \lambda_{\text{exp}}\mathcal{L}_{\text{exp}}(\theta). \tag{25}$$

**Assumption F.11** (Log-domain factorization gaps). There exists a decomposition $g = (g^{(1)}, \ldots, g^{(m)})$ and $x = (x^{(1)}, \ldots, x^{(m)})$ and candidate factors $\{\rho_i^\mu(\cdot \mid x^{(i)})\}_{i=1}^m$ and $\{p_{\mathrm{neg},i}\}_{i=1}^m$ such that the log-densities admit bounded residuals:

$$\Delta_\rho(x,g) := \log \rho^\mu(g \mid x) - \sum_{i=1}^m \log \rho_i^\mu(g^{(i)} \mid x^{(i)}), \tag{26}$$

$$\Delta_{\mathrm{neg}}(g) := \log p_{\mathrm{neg}}(g) - \sum_{i=1}^m \log p_{\mathrm{neg},i}(g^{(i)}), \tag{27}$$

and there exist constants $\varepsilon_\rho, \varepsilon_{\mathrm{neg}} \geq 0$ such that

$$|\Delta_\rho(x,g)| \leq \varepsilon_\rho \text{ and } |\Delta_{\mathrm{neg}}(g)| \leq \varepsilon_{\mathrm{neg}} \tag{28}$$

on the support of the training pairs $(x,g)$.

**Theorem F.12** (Additive decomposition with explicit approximation error). *Under Assumption F.11 and the conditions of Theorem 4.1, there exists a function $c(x)$ independent of $g$ such that any population-optimal score satisfies*

$$f^*(x,g) = \sum_{i=1}^m \Big( \log \rho_i^\mu(g^{(i)} \mid x^{(i)}) - \log p_{\mathrm{neg},i}(g^{(i)}) \Big) +$$
$$c(x) \; + \; \delta(x,g), \tag{29}$$

*where the residual $\delta(x,g)$ obeys the uniform bound*

$$|\delta(x,g)| \leq \varepsilon_\rho + \varepsilon_{\mathrm{neg}}. \tag{30}$$

*Proof.* By Theorem 4.1, the population-optimal score has the form

$$f^*(x,g) = \log \rho^\mu(g \mid x) - \log p_{\mathrm{neg}}(g) + c(x),$$

for some $c(x)$ independent of $g$. Add and subtract the factorized log terms:

$$f^*(x,g) = \Big( \sum_{i=1}^m \log \rho_i^\mu(g^{(i)} \mid x^{(i)}) + \Delta_\rho(x,g) \Big) - \Big( \sum_{i=1}^m \log p_{\mathrm{neg},i}(g^{(i)}) + \Delta_{\mathrm{neg}}(g) \Big) + c(x)$$

$$= \sum_{i=1}^m \Big( \log \rho_i^\mu(g^{(i)} \mid x^{(i)}) - \log p_{\mathrm{neg},i}(g^{(i)}) \Big) + c(x) + \underbrace{\big(\Delta_\rho(x,g) - \Delta_{\mathrm{neg}}(g)\big)}_{=: \, \delta(x,g)}.$$

This proves the decomposition (29). Under Assumption F.11, we have $|\Delta_\rho(x,g)| \leq \varepsilon_\rho$ and $|\Delta_{\mathrm{neg}}(g)| \leq \varepsilon_{\mathrm{neg}}$ on the support of training pairs, hence by the triangle inequality

$$|\delta(x,g)| = |\Delta_\rho(x,g) - \Delta_{\mathrm{neg}}(g)| \leq |\Delta_\rho(x,g)| + |\Delta_{\mathrm{neg}}(g)| \leq \varepsilon_\rho + \varepsilon_{\mathrm{neg}}.$$

$\square$

*Remark* F.13 (Interpretation and a design knob). Assumption F.11 is intentionally explicit: it isolates *why* object-centric additivity can fail in manipulation. The residual $\Delta_\rho$ captures inter-object coupling and any information loss induced by the object-centric projection $x \mapsto (x^{(i)})$; $\Delta_{\mathrm{neg}}$ captures non-factorized negative sampling. In particular, if negatives are constructed to factorize across objects (so that $p_{\mathrm{neg}}(g) = \prod_i p_{\mathrm{neg},i}(g^{(i)})$), then $\varepsilon_{\mathrm{neg}} = 0$ and the decomposition error is controlled purely by the dynamics/representation coupling term $\varepsilon_\rho$.

*Remark* F.14 (Expected factorization gap (KL / multi-information)). If one chooses $\rho_i^\mu(\cdot \mid x)$ such that $\prod_i \rho_i^\mu(\cdot \mid x)$ forms a product distribution over $\mathcal{G}$ for each $x$, then $\mathbb{E}_{g \sim \rho^\mu(\cdot \mid x)}[\Delta_\rho(x,g)] = \mathrm{KL}(\rho^\mu(\cdot \mid x) \| \prod_i \rho_i^\mu(\cdot \mid x^{(i)}))$. Thus, $\Delta_\rho$ can be interpreted as a conditional multi-information term measuring inter-object coupling and information loss from the object-centric projection. This suggests a principled diagnostic: estimate the KL gap (or a lower bound) to quantify when additive object-wise critics are justified.

Assumption F.11 is violated in manipulation due to contacts, occlusions, and coupled dynamics. The value of the object-centric architecture is therefore not that factorization is exact, but that attention over object tokens can approximate sparse interaction graphs (a structured critic), potentially improving credit assignment relative to monolithic global embeddings.

## F.7. DreamGen (video-only) augmentation

Let $\widetilde{\mathcal{D}}$ denote synthetic videos generated by a world model (e.g., DreamGen (Jang et al., 2025)), which provide state sequences but no actions. Training the implicit objective on $\widetilde{\mathcal{D}}$ corresponds to learning a goal-occupancy under a different rollout distribution $\tilde{\rho}$.

**Proposition F.15** (Mixture bias from synthetic rollouts). *If $\mathcal{L}_{\mathrm{imp}}$ is trained on a mixture of real demonstrations and action-free synthetic rollouts with mixing weight $\alpha \in [0,1]$, then the learned score corresponds to $f_\theta(x_t, g) = \log \rho_{\mathrm{mix}}(g \mid x_t) - \log p_{\mathrm{neg}}(g) + c(x_t)$, where $\rho_{\mathrm{mix}} = (1-\alpha)\rho^\mu + \alpha\tilde{\rho}$ and $\tilde{\rho}$ is the discounted occupancy induced by the synthetic rollouts.*

*Proof.* Let $p_{\mathrm{real}}^+(g \mid x) = \rho^\mu(g \mid x)$ and $p_{\mathrm{syn}}^+(g \mid x) = \tilde{\rho}(g \mid x)$. If each positive is drawn from the real source w.p. $1 - \alpha$ and from the

synthetic source w.p. $\alpha$, then by the law of total probability,

$$p_{\mathrm{mix}}^{+}(g \mid x) = (1 - \alpha)\rho^{\mu}(g \mid x) + \alpha\tilde{\rho}(g \mid x) =: \rho_{\mathrm{mix}}(g \mid x).$$

Applying Theorem 4.1 with $p^{+} = p_{\mathrm{mix}}^{+}$ yields

$$f_\theta(x, g) = \log \rho_{\mathrm{mix}}(g \mid x) - \log p_{\mathrm{neg}}(g) + c(x),$$

which proves the claim. $\square$

**Theorem F.16** (A simple robustness bound (relative error regime)). *Assume $\alpha\epsilon < 1$ and $\tilde{\rho}(g \mid x)$ satisfies a uniform relative-error bound $(1 - \epsilon)\rho^{\mu}(g \mid x) \le \tilde{\rho}(g \mid x) \le (1 + \epsilon)\rho^{\mu}(g \mid x)$ for all $(x, g)$. Then the mixture score bias is bounded as*

$$\big| \log \rho_{\mathrm{mix}}(g \mid x) - \log \rho^{\mu}(g \mid x) \big| \le \log\big(1 + \alpha\epsilon\big) - \log\big(1 - \alpha\epsilon\big). \tag{31}$$

*Proof.* Assume $(1 - \epsilon)\rho^{\mu}(g \mid x) \le \tilde{\rho}(g \mid x) \le (1 + \epsilon)\rho^{\mu}(g \mid x)$ for all $(x, g)$. Then the mixture occupancy $\rho_{\mathrm{mix}} = (1 - \alpha)\rho^{\mu} + \alpha\tilde{\rho}$ satisfies

$$(1 - \alpha)\rho^{\mu} + \alpha(1 - \epsilon)\rho^{\mu} \le \rho_{\mathrm{mix}} \le (1 - \alpha)\rho^{\mu} + \alpha(1 + \epsilon)\rho^{\mu},$$

i.e.,

$$(1 - \alpha\epsilon)\rho^{\mu}(g \mid x) \le \rho_{\mathrm{mix}}(g \mid x) \le (1 + \alpha\epsilon)\rho^{\mu}(g \mid x).$$

Assuming $\alpha\epsilon < 1$ (so the lower bound is positive), taking logs gives

$$\log(1 - \alpha\epsilon) \le \log \rho_{\mathrm{mix}}(g \mid x) - \log \rho^{\mu}(g \mid x) \le \log(1 + \alpha\epsilon).$$

Therefore,

$$\big| \log \rho_{\mathrm{mix}}(g \mid x) - \log \rho^{\mu}(g \mid x) \big| \le \max\{-\log(1 - \alpha\epsilon), \log(1 + \alpha\epsilon)\}.$$

Finally, note that $\max\{-\log(1 - \alpha\epsilon), \log(1 + \alpha\epsilon)\} \le \log(1 + \alpha\epsilon) - \log(1 - \alpha\epsilon)$, which yields the (looser) bound stated in the theorem. $\square$

*Remark* F.17 (Critical caveat: hallucinations destroy uniform bounds). Theorem F.16 requires a strong uniform approximation; in practice video generators can hallucinate (Jang et al., 2025), especially in low-data regimes. This motivates conservative weighting schemes that downweight synthetic positives unless they are validated by a discriminator or consistency check. Such conservatism is analogous in spirit to offline RL methods that penalize out-of-distribution value estimates, but extending those guarantees to representation-level occupancy estimation remains open.

In the video-only setting, the implicit objective is applied with state-only context $x = s$, hence estimating $\rho(g \mid s)$; learning $\rho(g \mid s, a)$ would require pseudo-actions (IDM/IGM/LAPA).

### F.8. Universal evaluation and multi-positive futures

**Proposition F.18** (Universal evaluation from occupancy). *Let $u : \mathcal{G} \to \mathbb{R}$ be bounded, and define the reward $r_u(s, a, s') := u(h(s')) = u(g_{t+1})$. Let $Q_u^\pi(s, a)$ denote the standard discounted action-value function under $\pi$ for reward $r_u$. Then*

$$Q_u^\pi(s, a) \;=\; \frac{1}{1 - \gamma} \, \mathbb{E}_{g \sim \rho^\pi(\cdot \mid s, a)}\big[u(g)\big]. \tag{32}$$

*Similarly, $V_u^\pi(s) = \frac{1}{1-\gamma}\mathbb{E}_{g \sim \rho^\pi(\cdot \mid s)}[u(g)]$.*

*Proof.* By definition of $Q_u^\pi$ with reward $r_u(s_t, a_t, s_{t+1}) = u(g_{t+1})$,

$$Q_u^\pi(s, a) = \mathbb{E}_\pi\left[\sum_{t=0}^\infty \gamma^t u(g_{t+1}) \,\middle|\, s_0 = s, a_0 = a\right].$$

On the other hand,

$$\begin{aligned}
\mathbb{E}_{g \sim \rho^\pi(\cdot \mid s, a)}[u(g)] &= \sum_{g \in \mathcal{G}} u(g)\,(1 - \gamma)\sum_{k=1}^\infty \gamma^{k-1} \Pr_\pi(g_k = g \mid s_0 = s, a_0 = a) \\
&= (1 - \gamma)\sum_{k=1}^\infty \gamma^{k-1} \mathbb{E}_\pi[u(g_k) \mid s_0 = s, a_0 = a] \\
&= (1 - \gamma)\sum_{t=0}^\infty \gamma^t \mathbb{E}_\pi[u(g_{t+1}) \mid s_0 = s, a_0 = a] \\
&= (1 - \gamma)\, Q_u^\pi(s, a),
\end{aligned}$$

which implies $Q_u^\pi(s, a) = \frac{1}{1-\gamma}\mathbb{E}_{g \sim \rho^\pi(\cdot \mid s, a)}[u(g)]$. The state-value case follows by marginalizing over $a \sim \pi(\cdot \mid s)$. $\square$

*Remark* F.19 (Why this matters). Proposition F.18 shows $\rho^\pi(\cdot \mid s, a)$ is a *universal critic* for any reward that depends on future goal variables (Dayan, 1993; Schaul et al., 2015). The sparse goal-reaching case in Proposition F.2 is recovered (up to the $(1-\gamma)$ normalization) by choosing $u(\tilde{g}) = \mathbf{1}\{\tilde{g} = g\}$; equivalently, choosing $u(\tilde{g}) = (1-\gamma)\mathbf{1}\{\tilde{g} = g\}$ matches the reward in Proposition F.2 exactly.

**Proposition F.20** (Multi-positive (multi-frame) sampling learns a mixture occupancy). *Let $\nu$ be a distribution over horizons $\{1, 2, \dots\}$. Define the induced positive distribution*

$$\rho_\nu^\mu(g \mid x) := \sum_{k=1}^\infty \nu(k) \Pr_\mu(g_{t+k} = g \mid x), \tag{33}$$

*where $x$ is the chosen context (either $s$ or $(s, a)$). If the implicit contrastive objective samples $k \sim \nu$ and uses $g = g_{t+k}$ as the positive, then the population-optimal InfoNCE score satisfies*

$$f_\theta(x, g) = \log \rho_\nu^\mu(g \mid x) - \log p_{\text{neg}}(g) + c(x), \tag{34}$$

*for some $c(x)$ independent of $g$. In particular, if $\nu = \text{Geom}(1 - \gamma)$ on $\{1, 2, \dots\}$, then $\rho_\nu^\mu = \rho^\mu$ in Definition F.1.*

*Proof.* Under the stated sampling scheme, the positive conditional distribution is exactly $p^+(g \mid x) = \rho_\nu^\mu(g \mid x)$ by the law of total probability over $k \sim \nu$. Applying Theorem 4.1 with this $p^+$ yields $f_\theta(x, g) = \log \rho_\nu^\mu(g \mid x) - \log p_{\text{neg}}(g) + c(x)$. For geometric $\nu$ on $\{1, 2, \dots\}$ with $\nu(k) = (1 - \gamma)\gamma^{k-1}$, we have $\rho_\nu^\mu = \rho^\mu$ by Definition F.1. $\square$

*Remark* F.21 (Interpreting "multi-frame" alignment). Sampling multiple future frames (or multiple offsets) corresponds to learning $\log \rho_\nu^\mu$ for a *mixture* over horizons (Oord et al., 2018). Geometric $\nu$ is the special choice that recovers a discounted Bellman fixed point.

**Summary.** Under geometric future sampling, the implicit future-alignment objective is equivalent (at the population optimum) to estimating the log density ratio of a goal-conditioned discounted occupancy (a value-like reachability quantity). In the state-conditioned setting used by FOCA, this corresponds to the state occupancy/value $\rho^\mu(g \mid s)$; an action-conditioned variant would instead recover $\rho^\mu(g \mid s, a)$ and a more direct $Q$-function interpretation for sparse goal-reaching rewards (Proposition F.2). Explicit object-centric prediction instead summarizes this occupancy via regression (Proposition 4.2), estimating a conditional mean that is directly useful for control. Their combination can be interpreted as a hybrid estimator with complementary failure modes; however, claims about policy improvement beyond the dataset behavior require additional offline RL machinery and careful distribution-shift analysis, especially when incorporating synthetic rollouts.

