# OpenReview forum: "FOCA: Future-Oriented Conditioning for Data-Efficient Vision-Language-Action Adaptation"
_ICML.cc/2026/Conference — ICML 2026 regular_

### Official Review · Reviewer_hVeA · 2026-03-02

**Soundness:** 2
**Presentation:** 2
**Significance:** 3
**Originality:** 2
**Overall Recommendation:** 4
**Confidence:** 5

**Summary:**

This paper studies the limitations of current vision language action models under few shot imitation learning and shows that their performance degrades sharply as the number of demonstrations decreases. To address this, the authors propose FOCA, a future oriented conditioning framework that adapts VLA models by explicitly modeling task grounded future interaction representations while implicitly aligning to future goal observations. The method enables long horizon reasoning in latent space without requiring pixel level future prediction and supports additional action free co training using synthetic videos from video world models. FOCA can be interpreted as learning a future conditioned value like representation that guides action generation. Extensive evaluations on simulation benchmarks and real world robotic tasks demonstrate substantial improvements in data efficiency and adaptation performance, establishing new state of the art results for few shot VLA adaptation.

**Compliance With Llm Reviewing Policy:**

Affirmed.

**Final Justification:**

The rebuttals addressed your main concerns.

**Key Questions For Authors:**

1. Detailed analysis on the computational overhead (in terms of flops or GPU hours).
2. [Important, would raise the score] Clarify the uniqueness of the few-shot stress test for VLA adaptation, or just remove it from the main contributions since the other contributions are sufficient.
3. Real world evaluations can further adds on the effectiveness of the proposed method.

**Limitations:**

yes

**Strengths And Weaknesses:**

Soundness
- The proposed explicit and implicit future alignment are empirically validated, and the implicit future alignment is theoretically analysed by drawing inspiration from contrastive learning. However, since extra video world model is required in stage 1, the computational overhead of  proposed method is doubtful, and further analysis should be provided.
- Since the few-shot stress test for VLA adaptation is claimed as unique contribution of this paper, but the boundary with previous works is unclear. For test setup in Q1 seems plain and widely-adopted.
Presentation.
- The paper is clearly written and easy to follow, and the figures are well-drawn to make readers to better understand the paper.
Significane.
- Few shot adaptation for VLA is an concrete problem for current stage of embodied intelligence before the era of zero-shot generalisation.
Meanwhile, data-efficient learning is a long-standing problem and has unique practical meaning for robotic manipulations.
Originality.
- The origins of the method can be interpreted as combination of v-jepa and contrastive learning objective (infoNCE), however, it indeeds solves the unique problem, data-efficient VLA adaptation.

---

> ### Author Rebuttal · Authors · 2026-03-31
>
> We thank the reviewer for the thoughtful feedback and for recognizing the strengths of our explicit-implicit design, its empirical support and theoretical grounding, as well as the clarity of our presentation. We also appreciate the acknowledgment of our contribution to the few-shot VLA adaptation. In this rebuttal, we address key concerns on DreamGen-related runtime, clarify the few-shot stress test, and provide additional real-world robot analysis.
>
> **Q1.  Analysis of the computational overhead in terms of GPU hours with DreamGen.**
>
> A1. We thank the reviewer for raising this important concern regarding computational overhead.
>
> We agree that incorporating a video world model (VWM) introduces additional offline cost, and we appreciate the opportunity to clarify this. Importantly, **the use of VWM in FOCA is optional**; our primary setting involves a single-stage fine-tuning procedure on pretrained VLA models, with computational cost comparable to standard adaptation (Appendix Sec. D.3).
>
> When a Video World Model (VWM) (e.g., DreamGen) is used, the training can be decomposed into stages as shown in Table 12 in the https://anonymous.4open.science/r/FOCA-V.
>
> As demonstrated in Table 12, the additional overhead mainly comes from (i) VWM fine-tuning (\~28 GPU hours depending on hardware) and (ii) FOCA co-training with synthetic data (\~15 GPU hours). The remaining stages are comparable to standard VLA fine-tuning (\~18 GPU hours). Importantly, this training structure is **consistent with prior pipelines that incorporate video world models (e.g, Gr00t)**, which typically involve (i) VWM fine-tuning, (ii) additional psuedo-action generation from generated videos, and (iii) policy fine-tuning. In our case, FOCA replaces the pseudo-action generation stage (e.g., IDM in prior works such as Gr00t) with **action-free supervision**, rather than introducing an extra stage.
>
> We emphasize that **this cost is incurred only once offline and does not affect deployment**. In contrast, inference cost remains unchanged, as FOCA does not rely on VWM at test time and only introduces a small number of tokens.
>
> Crucially, this **additional cost leads to significant gains in data efficiency**. For example, FOCA with VWM using only 40% real demonstrations achieves performance (95.7 in average) comparable to $\pi_0$ trained on full data (94.6 in average). This highlights a favorable trade-off: modest offline compute enables leveraging supervisory signals from synthetic videos that are otherwise difficult to obtain. We will include a clearer breakdown of GPU hours and this trade-off in the final version.
>
> **Q2. The boundary of “few-shot stress test for VLA adaptation” to prior work is unclear.**
>
> A2. We thank the reviewer for raising this important point. While a few prior works consider low-data regimes in VLA, **they primarily evaluate their own methods in isolation**. In contrast, our few-shot stress test differs in both scope and objective.
>
> - First, we conduct a systematic evaluation across a diverse set of SOTA VLA models (e.g., $\pi_0$, Gr00t-N1.5, E-01, Smol-VLA), analyzing their behavior under low-data regimes. Prior works did not study how these models degrade or generalize under strict data constraints.
>
> - Second, our “few-shot stress test” is not merely a data subsampling setup; it is designed to probe data efficiency as a core capability. Specifically, **we ask whether a method trained with limited data** (e.g., 10%-40%) **can match or approach the performance of the same backbone trained on full data**. This shifts the focus from scaling performance to **closing the gap under data scarcity**, which we believe is underexplored and practically important.
>
> We will clarify this distinction more explicitly in the final version.
>
> **Q3. Real-world evaluations can further add to the effectiveness of the proposed method.**
>
> A3. We thank the reviewer for the suggestion. Indeed, **we have already included real-world robot experiments, as shown in Fig. 3(a)**, covering tasks such as Tie Shoelaces, Set Table, and Open Bag, with detailed setups and results provided in Figure 6 of the Appendix. For the convenience of the reviewers, we have also reattached these results via Figure 12 in the anonymous link. We will further improve the presentation to make the real-world experiments clearer and easier to follow.

---

> > ### Author Rebuttal · Reviewer_hVeA · 2026-04-01
> >
> > Thank you for the rebuttal. The clarification on DreamGen overhead and the positioning of the few-shot stress test (2nd bullet point) mostly address my concerns. However, Figure 6 still appears somewhat vague and visually over-smoothed. Overall, the rebuttal is largely satisfactory, but the real-world evidence would be more convincing with clearer visualization and more explicit discussion of limitations.

---

> > > ### Author Response · Authors · 2026-04-03
> > >
> > > We thank the reviewer for the helpful feedback and are glad that our earlier clarifications addressed the concerns regarding (i) DreamGen overhead and (ii) the positioning of the few-shot stress test.
> > >
> > > We agree that Figure 6 (real-world setup) could be clearer. The current figure prioritizes compactness across multiple tasks, which can obscure fine-grained details (e.g., gripper–object interactions) and intermediate dynamics. To improve clarity, we provide higher-resolution visualizations, additional intermediate frames, and zoomed-in views (including start and end states) in the section ***Example rollouts from ALOHA real-world robot*** (https://anonymous.4open.science/r/FOCA-V). We hope these additions better convey both the setup and the challenges of the tasks, and we will incorporate them into the final version.
> > >
> > > Regarding limitations in real-world deployment, we observe several challenges. First, the model might struggle with tasks requiring precise 3D understanding (e.g., grasping a flat plate or manipulating a keychain), where 2D visual input is often insufficient. Second, for long-horizon tasks such as shoe tying, failures in intermediate steps are usually difficult to recover from, as the model lacks an explicit memory or state-tracking mechanism. These observations suggest promising directions for future work, including incorporating richer 3D representations (e.g., depth or spatial heatmaps), contact-point awareness, and introducing memory mechanisms for long-horizon reasoning. We thank the reviewer again for the suggestion and will incorporate these discussions into the revised version. If this response can address remaining concerns, we would appreciate it if the Reviewer could reconsider your rating.

---

### Official Review · Reviewer_PPfk · 2026-03-12

**Soundness:** 3
**Presentation:** 3
**Significance:** 4
**Originality:** 3
**Overall Recommendation:** 5
**Confidence:** 3

**Summary:**

This paper considers the problem of few-shot imitation learning (IL). The authors evaluate existing state-of-the-art VLA models, introduce a future-oriented conditioning approach to improve few-shot IL, FOCA, and verify their method both in simulation and in on real robots.

Unlike prior works that use future states, FOCA operates only in the latent embedding space. FOCA fine-tunes the VLM component of a VLA with tokens based on future observations. These tokens are explicit future latents, which predict the representation around the gripper and task-relevant objects, and implicit latents, which encode long-horizon task structure.

**Compliance With Llm Reviewing Policy:**

Affirmed.

**Final Justification:**

My concerns have been addressed in the rebuttal. There are also no concerns in the other reviews that have not been adequately addressed in the rebuttal.

**Key Questions For Authors:**

The paper mentioned FLARE, which is a method used during pre-training, but it is not clear to me why it couldn't be used during finetuning and is not present as a baseline.

**Limitations:**

A minor limitation I want to note is that FOCA does assume a certain breed of VLA that can be composed into VLM and action expert and is thus not architecture-agnostic. One could, for example, imagine a fully end-to-end system.

**Strengths And Weaknesses:**

Strengths
- The few-shot imitation learning problem is well justified, and future-conditioning achieves decent improvements over other approaches
- The ablations answer important questions about comparisons to other contrastive strategies and pixel-level predictions
- Impressively, the authors show that FOCA also allows the use of actionless video data to improve fine-tuning of the policy

Weaknesses
- I find some of the main figures hard to navigate. Figure 3 goes from c to a, b, d, e. Figures 2 and 3 contain both tables and Figures, but Table 1's caption is at the top. I'm not sure if this will fix it, but maybe keeping Figures, Tables, and pictures separate would help.
- As far as I can tell, "10," "Goal," "Object," and "Spatial" are neither explained in the text nor in the captions of the experimental results. From the Appendix, I understand that those are LIBERO-10, LIBERO-Goal, etc., but I think this should be clarified.
- The bolding is wrong in Figure 2 (a) for "10," "Goal," and "Spatial."

---

> ### Author Rebuttal · Authors · 2026-03-31
>
> We thank the reviewer for the careful and thoughtful feedback. We appreciate the recognition of our well-motivated few-shot setting, the effectiveness of our future-oriented conditioning, and the strength of our empirical analysis, including ablations. We are also grateful for highlighting FOCA’s practical ability to leverage action-free video data. We address the questions below.
>
> **Q1. Fix presentation issues in tables and figures.**
>
> A1. We thank the reviewer for these helpful suggestions and apologize for the confusion caused by the presentation. We will revise the figures to ensure a clear and consistent ordering (e.g., a → b → c → d → e) and improve the layout by separating figures and tables where appropriate, as well as standardizing caption placement.
>
> We will also clarify the terminology in both the main text and captions, explicitly stating that “10,” “Goal,” “Object,” and “Spatial” correspond to LIBERO-10, LIBERO-Goal, LIBERO-Object, and LIBERO-Spatial, respectively.
>
> Finally, we will correct the formatting issue in Fig. 2(a), including the incorrect bolding, to ensure accuracy and consistency.
>
> **Q2. Applicability of FLARE in fine-tuning and baseline comparison.**
>
> A2. We thank the reviewer for the question. **FLARE is used in Gr00t-N1.5 pre-training, but its implementation is not publicly available**, making faithful reproduction for fine-tuning unclear. Additionally, the released Gr00t code for the fine-tuning phase does not include FLARE-related tokens, suggesting it is not intended for this stage.
>
> However, motivated by the reviewer’s suggestion, we re-implement and integrate the FLARE mechanism into the same Pi-zero backbone on 40% LIBERO and the Gr00t-N1.5 backbone on the 100% RoboCasa setting during fine-tuning, following its core idea from [1], where the Q-Former is trained from scratch to pool VLM’s features due to the lack of pre-trained weights. We then evaluate it as a baseline against FOCA. The results shown in Tables 10 and 11 in the https://anonymous.4open.science/r/FOCA-V indicate that FOCA still surpasses this variant, suggesting that the gains primarily stem from our proposed future-oriented conditioning design rather than from future prediction signals alone.
>
> [1] Zheng, Ruijie, et al. "Flare: Robot learning with implicit world modeling." CoRL 2025.
>
> **Q3. Current applicability of FOCA to modular VLAs.**
>
> A3. We thank the reviewer for pointing out this perspective. FOCA currently applies to modular VLA architectures where a vision-language component and an action expert can be explicitly identified, reflecting the dominant design pattern in many recent strong VLAs such as $\pi_0$ and Gr00t, which enables integration with strong existing systems with minimal modification. We will update this perspective into the limitation section and discuss extension to fully end-to-end VLA architectures as a future direction.

---

> > ### Author Rebuttal · Reviewer_PPfk · 2026-04-03
> >
> > I thank the authors for their clarifications. My concerns have been addressed.
> > There are also no concerns in the other reviews that have not been adequately addressed in the rebuttal.
> >
> > Good work!

---

> > > ### Author Response · Authors · 2026-04-08
> > >
> > > We thank you Reviewer for your review of our rebuttal, and we are pleased that our response has addressed your concerns.

---

### Official Review · Reviewer_ijoy · 2026-03-12

**Soundness:** 3
**Presentation:** 3
**Significance:** 3
**Originality:** 3
**Overall Recommendation:** 4
**Confidence:** 3

**Summary:**

This work presents FOCA, which highlights using two types of auxiliary tokens (explicit and implicit tokens) to extract contextual understanding from the VLM backbone. The explicit tokens are projected to predict the visual embeddings of future interaction-focused areas, enhancing the model's ability to be properly aware of the future. Meanwhile, the implicit tokens are used to distinguish the embeddings from different tasks with an InfoNCE loss. During deployment, the explicit tokens, initialized as learnable parameters, are fixed, while the implicit tokens are dynamically generated.

**Compliance With Llm Reviewing Policy:**

Affirmed.

**Final Justification:**

This work presented a method that introduced two kinds of query tokens to introduce the future-oriented condition to VLA models.

During rebuttal,

1. My concern on the insight behind asymmetric initialization of query tokens has been resolved by explanations.
2. The author claims that my concerning design choices with probabilistic may introduce extra complexity, which is reasonable.
3. Extra experiments with DreamVLA + Pi0 backbone has been conducted for fair comparison.
4.  Explanation on disablement of explicit future prediction when training on DreamGen video has been demonstrated with the fact that synthetic videos may contain artifacts. Though reasonable, such disablement do harm the claim of the significance of proposed explicit tokens. Wish such drawback can be solved with future explorations.

Thus, I would like to remain my positive score of 4 (Weak Accept).

**Key Questions For Authors:**

1. On the asymmetric initialization of query tokens: Why are the explicit tokens parameterized as static learnable vectors rather than being dynamically extracted from current visual embeddings like the implicit tokens? Vice versa, why aren't implicit tokens initialized as static learnable parameters? The rationale behind this asymmetric design choice is unclear, and the paper lacks ablation studies to justify why such specific initialization methods are optimal for their respective tasks.
  2. On the parameterization of explicit tokens during deployment: Why are the explicit tokens simply fixed as static learned parameters during inference? Such rigid modeling may impair the model's ability to generalize to out-of-distribution (OOD) scenarios, as potentially implied by the performance bottlenecks in extreme few-shot settings. Would it be more principled to model these tokens probabilistically? For instance, similar to the latent variable design in ACT, would it be more reasonable to parameterize these representations via the mean and variance of a Gaussian distribution during training, and subsequently set them to the prior mean (e.g., zero) during deployment to output the most generalized conditional behavior?
  3. On the synthetic video co-training: The paper highlights action-free learning from synthetic videos as a key contribution. However, the explicit future prediction module is disabled when training on DreamGen videos. Could the authors clarify reasons behind this design choice? Does it suggest that the explicit module is sensitive to hallucinations or temporal inconsistencies in generated videos? Discussing this would help readers better understand the applicability of the explicit module.
  4. On the baseline selection: While "future-oriented conditioning" is the core freature of proposed method, the main experiments primarily compare FOCA against vanilla base models and generic fine-tuning methods. It would strengthen the paper to include system-level future-oriented or goal-conditioned baselines (e.g., GCBC) in the main evaluation. This addition would help clearly attribute the performance gains to FOCA's specific architectural design, rather than the general advantage of accessing future information in few-shot settings.

**Limitations:**

yes

**Strengths And Weaknesses:**

Strengths:
  1. The paper is well-written with a timely motivation for few-shot VLA adaptation.
  2. Comprehensive evaluations are conducted across diverse simulation and real-world benchmarks with competitive results.
  3. Ablation studies effectively prove that both explicit and implicit auxiliary tokens collaboratively enhance performance.

  Weaknesses:
  1. The asymmetric token initialization (static learnable explicit tokens vs. dynamically extracted implicit tokens) lacks justification.
  2. Rigidly fixing explicit tokens as static parameters during deployment may impair out-of-distribution generalization, leaving probabilistic modeling (like ACT) unexplored.
  3. The main experiments lack system-level future-oriented baselines

---

> ### Author Rebuttal · Authors · 2026-03-31
>
> We thank the reviewer for the thoughtful feedback. We appreciate the recognition of our motivation for few-shot VLA adaptation, the clarity of the presentation, and the strength of our evaluation across simulation and real-world benchmarks, as well as the effectiveness of FOCA’s explicit and implicit design. We address the questions below.
>
> **Q1. What is the rationale behind the asymmetric initialization of explicit tokens (static learnable vectors) and implicit tokens (dynamically extracted features)?**
>
> A1. We thank the reviewer for this insightful question. The asymmetric initialization stems from the different intended roles of implicit and explicit tokens, but we agree that this design choice requires empirical validation.
>
> **Implicit tokens** are meant to capture the current visual context for alignment with future states. Initializing them dynamically from visual features ensures they reflect the scene at each timestep, which we found important for stable and meaningful alignment.
>
> **For explicit tokens**, our initial goal was to study their role as a compact set of predictive tokens for future latent representations, rather than to optimize their initialization strategy. As such, we adopted a simple design by initializing them as learnable parameters, leaving more sophisticated alternatives unexplored in the initial submission.
>
> **To better understand this design choice, we conducted additional ablations:** (i) initializing implicit tokens as random/static parameters, and (ii) initializing explicit tokens from visual features followed by a learnable projection. The results (Tables 6 and 7 in https://anonymous.4open.science/r/FOCA-V) show that dynamic initialization is particularly important for implicit tokens, while explicit tokens are less sensitive but still benefit from it. In fact, making both token types dynamic yields the best performance. We will include these results in the revised version to provide a more complete justification of the design.
>
> **Q2. Parameterization of explicit tokens, design choices with probabilistic.**
>
> A2. We thank the reviewer for this insightful suggestion. As discussed in Q1, our additional ablations show that FOCA remains effective and can further improve when explicit tokens are dynamically initialized from visual features, indicating that conditioning tokens on the current observation is a promising alternative version.
>
> We agree that probabilistic parameterizations (e.g., Gaussian latent tokens) are a favourable direction, as they could capture uncertainty in future prediction and potentially integrate well with frameworks such as ACT. However, at the same time, they might introduce non-trivial complexity (e.g., sampling, stability, and objective design), which would require careful treatment to evaluate fairly.
>
> Given that our primary goal is to isolate and study the impact of deterministic explicit/implicit token design and future-aware conditioning, we leave probabilistic extensions as future work.
>
> **Q3. Comparing with system-level future-oriented baselines.**
>
> A3. We thank the reviewer for this concern. In the current submission, we already compare against future-aware systems such as **DreamVLA, CoT-VLA, and Think-Act** (Fig. 2a). These methods leverage future information via dynamic-region forecasting, pixel-level subgoal generation, or intermediate reasoning. FOCA consistently outperforms them, indicating that the gains come from our latent, object-centric future token design rather than simply incorporating future information.
>
> In this rebuttal, we further investigate the contribution of future-oriented conditioning by **integrating DreamVLA-style techniques into the same Pi-zero backbone** and evaluating on both 40% and 100% scales of the LIBERO dataset. As shown in Tables 8 and 9 in the anonymized link, while dynamic future-aware conditioning provides some improvement, it still underperforms FOCA, indicating that the gains mainly come from our latent, object-centric future token design rather than simply incorporating future information.
>
> **Q4. Discussion on the disablement of the explicit future prediction module when training on DreamGen videos.**
>
> A4. We thank the reviewer for this insightful question. The explicit future prediction module relies on accurate, temporally consistent, and object-centric supervision to model fine-grained interaction dynamics. However, synthetic videos (e.g., DreamGen) may contain artifacts, such as spatial inaccuracies and temporal inconsistencies, which can corrupt these localized signals and render explicit prediction unreliable.
>
> Therefore, we disable the explicit module during synthetic co-training and instead rely on the implicit alignment objective, which operates at a higher-level representation and is more robust to such noise. Importantly, it does not require precise localization or action labels, making it better suited for leveraging large-scale synthetic data.

---

> > ### Author Rebuttal · Reviewer_ijoy · 2026-04-02
> >
> > I thank the authors for their explanations, which have successfully addressed my concerns.

---

> > > ### Author Response · Authors · 2026-04-03
> > >
> > > We thank you Reviewer for your review of our rebuttal, and we are pleased that our response has addressed your concerns.

---

### Official Review · Reviewer_g4xK · 2026-03-13

**Soundness:** 3
**Presentation:** 4
**Significance:** 3
**Originality:** 4
**Overall Recommendation:** 5
**Confidence:** 3

**Summary:**

This paper addresses the critical problem of data inefficiency in adapting pretrained VLA models to new tasks under few-shot imitation learning settings. The authors observe that standard behavioral cloning provides sparse supervision and fails to leverage the rich future information contained in demonstrations. To address this, they propose FOCA, a framework that injects future-oriented supervision into VLA adaptation. FOCA introduces two auxiliary objectives: (1) Explicit Future Prediction, which predicts future latent embeddings of task-relevant interaction regions, and (2) Implicit Future Alignment, a contrastive objective that aligns current representations with future goal observations. Crucially, the implicit objective requires no action labels, allowing the model to leverage synthetic videos from video world models for action-free co-training.

**Compliance With Llm Reviewing Policy:**

Affirmed.

**Final Justification:**

This paper presents a well-motivated and clearly explained future-oriented adaptation framework for VLA models that demonstrates strong empirical results, and the authors’ rebuttal effectively addressed my concerns, so I keep my positive assessment and original recommendation to accept (5).

**Key Questions For Authors:**

1. In the explicit prediction branch, what happens if the grounding model fails to identify the task-relevant objects or if the bounding box is inaccurate? Does the loss function or architecture include any mechanisms to filter out incorrect grounding signals?
2. In the real-robot experiments (Figure 3a), was the model pre-trained on synthetic videos generated by DreamGen, or was it trained solely on the limited real demonstrations using the explicit/implicit losses? If synthetic data was not used for real robots, why not?
3. What is the method's sensitivity to sampling distribution? The theoretical interpretation relies on sampling the future offset from a geometric distribution to approximate discounted occupancy. How sensitive is the performance to the choice of discount factor or the specific sampling strategy?

**Limitations:**

yes

**Strengths And Weaknesses:**

# Strengths
1. The paper identifies a practical and significant weakness in current VLA models which is often overlooked in evaluations using large demonstration budgets.
2. The combination of explicit latent prediction and implicit contrastive alignment is a novel and well-designed solution. It is a good insight that the implicit alignment objective can utilize action-free synthetic videos.
3. The empirical work is extensive, covering multiple simulation benchmarks and diverse real-robot embodiments.
# Weaknesses
1. The explicit prediction branch relies on an off-the-shelf grounding model  to identify object-centric bounding boxes. The paper does not thoroughly analyze how errors or failures in this grounding model affect the overall system, which introduces a potential point of failure in novel domains.
2. The implicit alignment objective assumes that the "future" frames sampled from demonstrations represent successful task progression. If the dataset contains low-quality or failed demonstrations (common in real-world robot data), aligning with these "negative" futures could misguide the policy.
3. The method introduces several new components (explicit/implicit tokens, token isolation masks, geometric sampling) and loss terms. This increases the implementation complexity and introduces hyperparameters that may require careful tuning.

---

> ### Author Rebuttal · Authors · 2026-03-31
>
> We thank the reviewer for the thoughtful feedback. We appreciate the recognition of data efficiency in few-shot VLA adaptation, the strengths of our explicit-implicit design and use of action-free synthetic videos, as well as the breadth of our evaluation across simulation and real-world settings. We address the questions below.
>
> **Q1. How sensitive is FOCA to errors from the off-the-shelf grounding model used for object-centric regions? Discuss mechanisms to mitigate or filter incorrect grounding signals during training?**
>
> A1. FOCA is reasonably robust to errors from the off-the-shelf grounding model, as **grounding is used only during training** to provide **auxiliary object-centric supervision** and is **not required at inference time**. The deployed policy operates purely on raw observations, language inputs, and learned tokens.
>
> To mitigate sensitivity to imperfect grounding during training, we expand predicted bounding boxes with a margin calibrated from a small set of manually inspected samples. This helps ensure that relevant interaction regions are still captured even when the boxes are misaligned or incomplete.
>
> At present, we do not explicitly filter or reweight noisy grounding signals, which we will acknowledge as a potential source of noise and a direction for future improvement. However, in practice, robustness is further supported by two factors: (i) grounding models can be improved via lightweight fine-tuning on a small set of demonstrations, and (ii) supervision is applied in latent space, making it less sensitive to precise spatial alignment and more tolerant to imperfect boxes.
>
> **Q2. How does noise in demonstrations affect the implicit alignment objective, which assumes that future frames reflect successful task progression?**
>
> A2. We agree that low-quality or failed demonstrations can negatively affect the implicit alignment objective, as aligning current representations with noisy or suboptimal “future” states may misguide the policy. To mitigate this issue in practice, we apply a **data filtering preprocessing step** (described in Sec. E of the Appendix) to remove low-quality or noisy samples before training, for both simulated and real-world datasets. This follows common practices, such as in OpenVLA [1]. As a result, future frames used for alignment more reliably reflect meaningful task progression, helping stabilize the training signal for the implicit objective.
>
> Additionally, the implicit objective operates at the representation level rather than enforcing exact pixel-level correspondence. While this does not eliminate the impact of noise, it makes the objective less sensitive to small visual inconsistencies and partial failures, as long as the overall trajectory still reflects task-relevant progress.
>
> [1] Kim, Moo Jin, et al. "OpenVLA: An open-source vision-language-action model." CoRL 2024.
>
> **Q3. FOCA might require careful optimization and tuning of hyperparameters. Analyze FOCA’s results w.r.t the time step sampling in the geometry perspective.**
>
> A3. We agree that FOCA introduces additional components and that hyperparameter choices (e.g., token numbers and sampling strategy) can affect performance. However, in our implementation, we showed that this effort is **tuned only on the LIBERO dataset** and then **kept fixed across other benchmarks**, including RoboCasa and real-world experiments. Despite this, FOCA consistently improves over baselines, suggesting it is **not overly sensitive** to these choices in practice, and does not require extensive retuning for new environments.
>
> Regarding the sampling strategy, geometric sampling is primarily motivated by the theoretical interpretation in Sec. 4.2, which corresponds to a discounted-occupancy formulation. Empirically, we observe that the implicit alignment module remains effective even under simpler alternatives (e.g., fixed-horizon variants). For example, in Fig. 3(e), using a **single future frame** or **multiple future frames** yields comparable performance under the 10% LIBERO setting, indicating robustness to the specific sampling design.
>
> **Q4. Why has DreamGen with FOCA on a real-robot setup not been included yet?**
>
> A4. We thank the reviewer for this suggestion. Evaluating FOCA on real-robot setups already involves substantial overhead, including hardware setup, camera calibration, and repeated real-world trials across multiple tasks. Integrating DreamGen further adds complexity in generating and validating synthetic data within each pipeline. Due to these combined efforts and the limited submission timeline, we were not able to include a full DreamGen+FOCA evaluation on real robots.
>
> To partially address this concern, we provide additional results on table setup tasks under the 40% and 100% demonstration setting **(Table 5, https://anonymous.4open.science/r/FOCA-V)**, where DreamGen further improves performance, indicating its potential to raise FOCA’s upper bound. We will update these ones in the revised version.

---

> > ### Author Rebuttal · Reviewer_g4xK · 2026-04-03
> >
> > Thank you for the detailed rebuttal and clarifications. The explanations and additional experiments help me better understand FOCA’s robustness to grounding errors, low-quality demonstrations, and hyperparameter choices. I maintain my positive assessment and original score.

---

> > > ### Author Response · Authors · 2026-04-08
> > >
> > > We thank you Reviewer for your review of our rebuttal, and we are pleased that our response has addressed your concerns.

---

### Decision · Program_Chairs · 2026-04-30

**Decision:**

Accept (regular)

**Comment:**

Reviewers broadly agreed that the paper addresses a well-motivated and practically important problem — data-efficient adaptation of VLA models under few-shot imitation learning — and that the proposed FOCA framework makes a meaningful contribution to this setting. Specific strengths highlighted across reviews include: the novel combination of explicit future latent prediction and implicit contrastive alignment; the practical and theoretically grounded insight that the implicit objective enables action-free co-training with synthetic videos; and the breadth of evaluation spanning multiple simulation benchmarks and diverse real-robot embodiments, which collectively support the paper's core claims.
Following a substantive rebuttal, three of four reviewers marked their concerns as fully resolved, and no reviewer lowered their score. The following specific issues should be addressed before the camera-ready version:
Explicit module and synthetic data compatibility. The explicit future prediction module is disabled during DreamGen co-training due to artifacts in synthetic videos. While the authors provide a reasonable explanation, this design choice limits the scope of one of the paper's claimed contributions and should be discussed more honestly in the main text as a known limitation, rather than deferred to a footnote or appendix.
Presentation quality. Reviewers identified concrete issues including inconsistent figure ordering, undefined benchmark abbreviations ("10," "Goal," "Object," "Spatial") in main-text figures, and incorrect bolding in Fig. 2(a). These should be corrected in the final version.
Real-world visualization. The real-robot figures were noted as visually over-smoothed and lacking sufficient detail (e.g., intermediate gripper states, failure cases). Higher-resolution visualizations and an explicit qualitative discussion of real-world failure modes — as partially provided in the rebuttal — should be incorporated into the paper.
Asymmetric token initialization. The additional ablations provided in the rebuttal (Tables 6 and 7 in the anonymous link) directly address this concern and should be included in the final paper to provide a complete empirical justification for the design.
Overall, the paper presents a technically solid and empirically well-supported contribution to few-shot VLA adaptation, and the rebuttal process meaningfully strengthened the submission. The paper is recommended for acceptance.